# The 2023 M$_w$ 7.8–7.7 Kahramanmaraş earthquakes were loosely slip-predictable
Ellis Vavra [1] ✉, Yuri Fialko[1], Fatih Bulut[2], Aslı Garagon[2], Sefa Yalvaç[3] & Cenk Yaltırak [4]

Understanding the behavior of large earthquakes over multiple seismic cycles is limited by short time spans of observations compared to recurrence intervals. Most of large instrumentally-recorded earthquakes have occurred on faults lacking well-documented histories of past events. The 2023 M$_w$ 7.8–7.7 Kahramanmaraş earthquake doublet is exceptional as it ruptured multiple segments of the East Anatolian Fault (EAF) system, where historical records of devastating earthquakes span over two millennia. Here, we use historical earthquake records, measurements of interseismic deformation, and published slip models of the 2023 events to evaluate the recurrence patterns of large earthquakes. We compare slip deficit that accrued on each fault segment since the respective penultimate events to the average coseismic slip of the 2023 doublet. We find that the coseismic slip equaled to or exceeded the accumulated slip deficit, suggesting that the slip-predictable recurrence model applies as a lower bound on strain release during the Kahramanmaraş earthquakes.

Forecasting the timing and size of future earthquakes is a fundamental objective of earthquake science. While several models for earthquake recurrence have been proposed, none have been shown to be universally applicable to natural earthquakes[1–3]. For a fault with a constant loading rate and an upper stress threshold for failure, earthquakes are expected to be "time-predictable," wherein the time until an event is determined by the size of the previous event and fault loading rate. In the "slip-predictable" case, where the fault has a lower bound on stress (e.g., due to residual friction), earthquakes are expected to release all accumulated strain since the preceding event. While these recurrence models are highly simplistic, idealized descriptions of earthquake cycle behavior, they attracted much attention due to their potential for assessing first order characteristics about the timing, frequency, and size of large earthquakes. Evaluating models of earthquake recurrence requires knowledge of (1) the timing, size, and location of previous earthquakes, and (2) the rates of tectonic strain accumulation over the intervening interseismic period(s). While the interseismic deformation is reasonably well constrained using modern geodetic techniques[4,5], information about past earthquakes is often lacking, incomplete, and/or subject to large uncertainties[6–8], hindering quantitative assessments of the earthquake recurrence models.

Previous studies evaluating earthquake recurrence models have produced mixed results, suggesting that neither the time- nor the slip-predictable model is universally applicable, irrespective of location, faulting style, and earthquake magnitude[1–3,9–15]. Some analogs of natural earthquakes, such as stick-slip events observed in laboratory experiments[16] and at

the glacier beds[9], were argued to be slip-predictable. Groups of repeating earthquakes along the San Andreas fault (SAF) system in California appear to be "loosely" slip-predictable, in that their magnitude scales (albeit imprecisely) with the interval since the preceding event[10,11]. However, regional catalogs of instrumentally recorded earthquakes spanning several decades (for instance, over 20 years of seismicity in Italy[12]) typically show no evidence for either slip- or time-predictable behavior. In some cases, long paleoseismic records (spanning more than several events) allow useful insights into earthquake recurrence on major fault systems[13–15]. Along the Alpine fault in New Zealand, the recurrence interval of large events appears to be quite regular (329 ± 68 yrs), but departs from classical recurrence models due to variations in slip per event[14]. On the Carrizo Plain segment of the central SAF, system-size events were found to be loosely slip-predictable[15]. In contrast, paleoseismic data at the Wrightwood site further to the south were interpreted as lacking any support for either time- or slip-predictable model[3]. In Türkiye, the central Gerede segment of the North Anatolian Fault (NAF) has exhibited quasi-regular, characteristic behavior over the last 1000 years that is consistent with the slip-predictable model[15].

It is possible that the efficacy of the time- or slip-predictable models may be scale-dependent. Small to moderate earthquakes may be governed by local heterogeneities in stress, including stress changes from other earthquakes, pore fluid pressure, and material properties, and thus exhibit more chaotic behavior—or at least appear to due to the challenges of probing the physical state of faults on sub-kilometer scales. On the other hand, large "system-size" (i.e., rupturing the entire seismogenic zone) events

[1]Institute of Geophysics and Planetary Physics, Scripps Institution of Oceanography, University of California, San Diego, La Jolla, CA, USA. [2]Geodesy Department, Boğaziçi University, Kandilli Observatory and Earthquake Research Institute, Istanbul, Türkiye. [3]Survey Engineering Department, Gümüşhane University, Faculty of Engineering, Gümüşhane, Türkiye. [4]Geology Department, Istanbul Technical University, Faculty of Mines, Istanbul, Türkiye. ✉e-mail: evavra@ucsd.edu

may be less susceptible to small-scale heterogeneities, and perhaps more sensitive to integral characteristics such as an average stress in the seismogenic zone and a long-term tectonic loading rate. If so, large events might be easier to correlate with a limited number of controlling parameters, compared to numerous small events that are affected by a variety of factors, most of which are currently observationally inaccessible. The challenge is that our knowledge of past major events is often insufficient to thoroughly examine potential patterns in seismic cycles on any given fault system.

The remarkable 2023 $M_W$ 7.8–7.7 Kahramanmaraş earthquakes in Türkiye, which are considered to be doublet due to their similar large magnitudes and occurrence in rapid succession (~9 hrs apart)[17,18], present a rare opportunity to assess the nature of earthquake recurrence in the case of major (M > 7) events thanks to: (1) well-documented coseismic deformation and detailed coseismic slip models[17,19–21], (2) geodetic and geologic estimates of interseismic slip rates[22–42], and (3) quality constraints on the time and size of prior earthquakes from historical records and paleoseismic data[22,43–47]. In addition, the extensive spatial extent of the ruptures allows us to compare recurrence behavior across different fault systems and segments. We find that coseismic slip on the main five fault segments ruptured during the 2023 sequence equaled to or exceeded the accumulated slip deficit since the preceding major earthquakes. The inferred slip overshoot on some sections of the $M_w$ 7.8 rupture may be due to its unusually large length, which resulted from cascading effects and rupture propagation across nominal segment barriers, tapping into residual stresses left behind by the penultimate M ~ 7 events. Our results demonstrate that for all fault segments that ruptured in the 2023 doublet, the average coseismic slip scales with the accumulated slip deficit, consistent with loose slip-predictability, in particular for segments that accrued the largest slip deficit.

## Coseismic rupture along the East Anatolian fault

**The 2023 $M_W$ 7.8–7.7 Kahramanmaraş earthquake doublet.** Türkiye is located within the Alpine-Himalayan orogenic belt, which is highly seismically active and capable of destructive earthquakes[48]. The country comprises a large portion of the Anatolian microplate, which is bounded by the North Anatolian and East Anatolian transform faults along its northern and eastern margins (NAF and EAF, see Fig. 1), respectively, and the Hellenic and Cyprian arcs to the south and southwest[49]. Together, these major plate boundary fault zones accommodate the westward movement of the Anatolian microplate with respect to Eurasia at a rate of several centimeters per year driven by the collisional tectonics in the east and extensional regime in the west[50]. The associated active deformation results in a major (M 7+) earthquake typically every 6–7 years in Türkiye[51].

On February 6, 2023 (01:17 GMT), a $M_w$ 7.8 earthquake occurred in Southern Türkiye, with an epicenter ~40 km south of the city Kahramanmaraş. It ruptured the entirety of the Amanos, Pazarcık, and Erkenek sections of the main strand of the EAF[17,29,52] with peak slip of over 8 m[17] (Fig. 2). Only about nine hours later, the $M_w$ 7.8 earthquake was followed by a $M_w$ 7.7 earthquake (10:24 GMT), initiating at ~95 km distance to the north from the epicenter of the $M_w$ 7.8 event and rupturing the Çardak and Göksun segments of the northern strand of the EAF system, which we refer to as the Gökun-Çardak Fault System (GCFS). The latter event alone would have been the largest earthquake in the region since the devastating August 17, 1668 M 7.9 earthquake on the NAF[43]; the total moment release of the doublet, equivalent to a $M_w$ 7.95 event[17], may have been the largest ever documented in Türkiye. The 2023 earthquake sequence had disastrous consequences, with widespread destruction of built infrastructure and estimated 50,500 deaths (April 14, 2023, Turkish Ministry of Interior). Due to the availability of dense instrumental networks and remote sensing observations, the 2023 Kahramanmaraş doublet has been exceptionally well recorded, providing unique insights into the seismogenic processes. To test existing models of earthquake recurrence[1,2], we begin by estimating the average slip on each fault segment that ruptured in the 2023 $M_w$ 7.8 and $M_w$ 7.7 events using finite fault models of the coseismic slip and aftershock locations.

**Cosesimic slip estimates.** Numerous studies have investigated the slip distribution of the two events in the 2023 doublet[17,19–21]. Choices in inversion methodology, model parameterization, and data curation, as well as model non-uniqueness, result in inherent variations in obtained distributed slip solutions[53–55] and can result in differences in models of the same earthquake rupture[56]. To make a robust assessment of the coseismic slip on faults that ruptured during the 2023 Kahramanmaraş doublet, we use several published finite fault models to quantify average slip along each fault segment and the associated uncertainty (see Table S1). Below, we outline our approach to integrating different slip models which involves recasting the modeled moment release on each segment onto a common fault geometry.

Since each model is discretized differently, we develop a unified framework for comparing the respective slip estimates. First, we adopt rupture geometries constrained by field mapping, inversions of space geodetic data, and aftershock locations (Fig. 1; see Supplementary Methods for details of aftershock catalog)[17,57,58]. We delineated each rupture trace into segments according to those defined by ref. 29, which are based on the geometry and jog structures of the EAF. The $M_w$ 7.8 event ruptured several segments along the main strand of the EAF, including the Amanos, Pazarcık, and Erkenek segments (Fig. 1c). The $M_w$ 7.7 event ruptured a portion of the Göksun segment, the entire mapped Çardak segment, and a previously unidentified

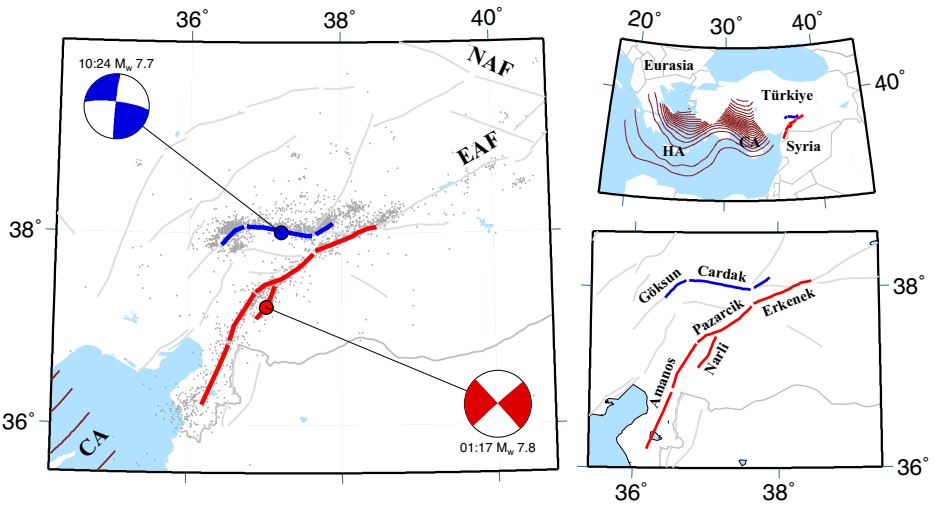

**Fig. 1 | The 2023 Kahramanmaraş Earthquakes.** In all panels, red and blue colors refer to the first and the second M 7+ events, color dots indicate the epicenters of each event, light gray lines denote seismically active faults in eastern Türkiye[57], dark gray lines are international borders, brown lines are 20 km contours for the Hellenic Arc (HA) and Cyprus Arc (CA)[49]. (left) Map of the 2023 Kahramanmaraş ruptures. Focal mechanisms for both the events are shown. Aftershocks locations[58] are shown as gray dots. (upper right) Map of the eastern Mediterranean region. (lower right) Ruptured fault segments of the East Anatolian Fault.

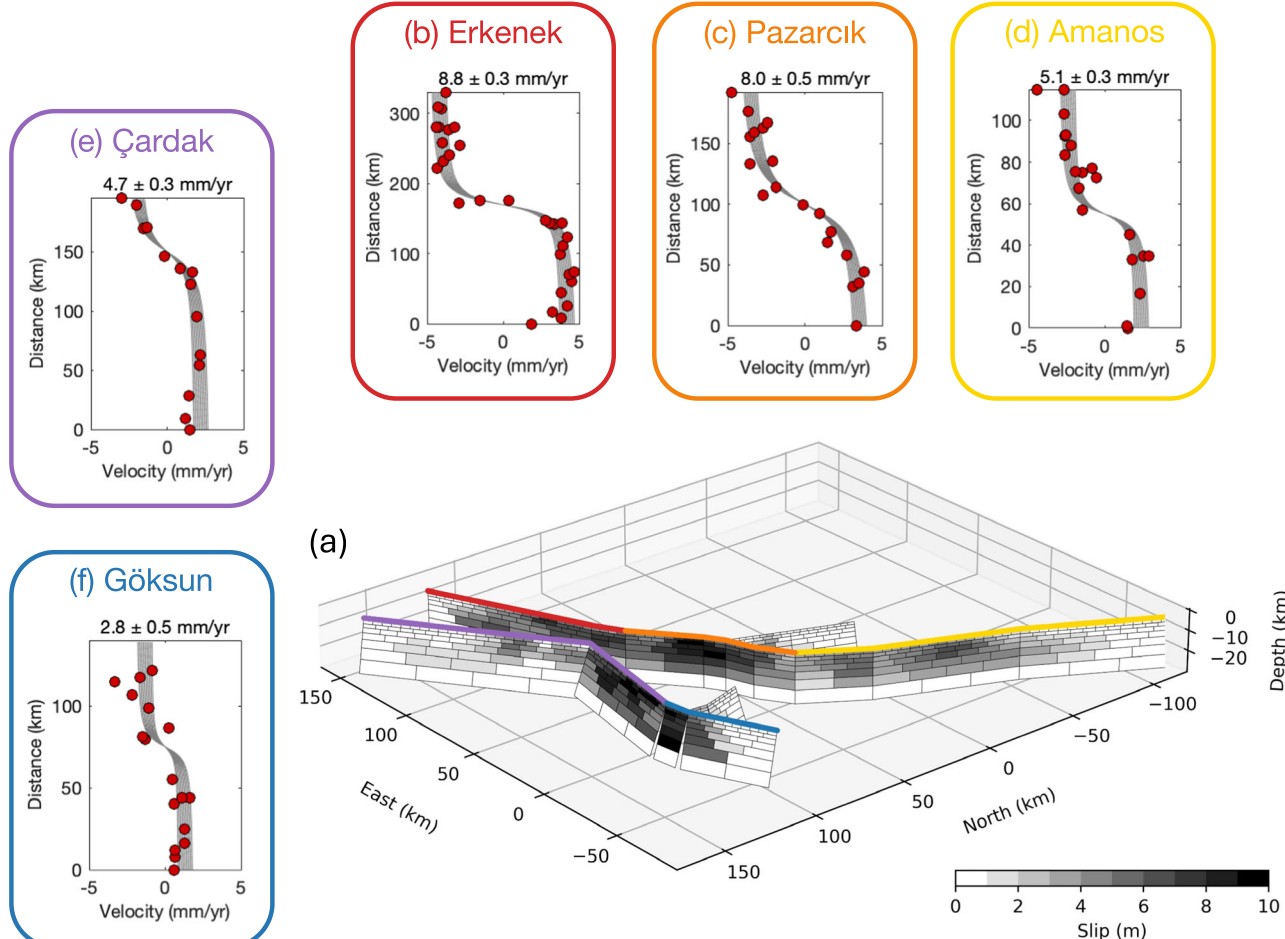

**Fig. 2 | Coseismic slip and interseismic deformation on faults ruptured by the 2023 earthquake doublet. a** Slip distribution along the February 6, 2023 rupture zones from ref. 17. The gray color scale for slip is saturated at 10 meters. **b–f** Pre-2023 interseismic GNSS velocities (red dots) and predictions of the best-fit dislocation model (gray lines) for major fault segments which ruptured during the 2023 earthquake sequence. The outline colors in (**b–f**) correspond to the surface traces shown in (**a**).

northeast extension of the Çardak segment parallel to the EAF (Fig. 1c). We note that we do not analyze slip from the Narli segment or splay fault off the Göksunsegment due to a lack of knowledge regarding their rupture history and geodetic slip rates.

We calculate the average coseismic slip on each fault segment. The average slip governs the seismic moment and is expected to be less variable between different finite fault models. Given a variable down-dip extent of the published slip models, and the fact that the inferred slip typically tapers off at a depth shallower than the bottom edge of the fault model, deeper parts of the model with effectively zero slip should be excluded from calculation of the average slip on a given segment. Correspondingly, we compute the average coseismic slip $\bar{s}$ by summing the moments from all slip patches in the segment $M_0^i$ and dividing by the effective segment area $A$ above the inferred locking depth $D_l$,

$$\bar{s} = \frac{\sum M_0^i}{A} = \frac{\sum M_0^i}{L D_l \sin\theta}, \quad (1)$$

where $L$, is the segment length, and $\theta$ is the segment dip angle. This way, we are conserving the moment release estimated in each individual inversion, but normalizing by the "seismogenic" thickness that is the same across all fault models.

We obtain an estimate of the average slip $\bar{s}$ along each fault segment in each model; we then compute the average and standard deviation over all models. Results are shown in Fig. 3.

The assumed effective segment area $A$ affects the average slip values, so care is needed to accurately constrain the geometric parameters. We use the mapped along-strike extent of each ruptured segment, based on the extent of surface ruptures, aftershock locations[58], and surface deformation[17], to separate the finite fault models info the aforementioned segments and estimate $L$. As all fault segments aside from the Pazarcık are bounded by only one adjacent segment (i.e., segments on which rupture terminated), the location of segment tips tends to vary from model-to-model. To conserve the total moment hosted by a segment regardless of its discretization, we do not truncate segments with tips located beyond the mapped rupture and use the same mapped $L$ for each model. The fault dip angles $\theta$ are primarily constrained by surface rupture and aftershock data[59] and are generally consistent amongst different finite fault models[17,19–21]. In particular, the EAF is interpreted to be near-vertical, and Çardak and Göksun faults have a moderate northward dip of 70–80o[17,19–21]. We assume dips of 90 degrees and 75 degrees for the EAF and GCFS, respectively. The primary potential source of ambiguity is in the fault locking depth $D_l$. To constrain the segment locking depths, we use the 95% cutoff depth of aftershocks and background seismicity (see Supplementary Methods). We find this depth to be in good agreement with the maximum depth extent of coseismic slip (see Methods; Fig. 3 and S1). We also estimate locking depths from inversions of interseismic GNSS data but find them to be less compatible with other available observations. Below, we discuss the influence of different locking depth choices on the inferred coseismic slip estimates.

**Fig. 3 | Along-strike variability in slip rates and coseismic slip on the East Anatolian fault.** Top row: slip rates along each segment of the EAF that ruptured in the 2023 Kahramanmaraş earthquakes. The dark red and blue lines show slip rates estimated in this study, while light red and blue lines and shading indicate the mean and standard deviation of rates from the studies listed in Table S2. Bottom row: coseismic slip estimates obtained from finite slip models of the 2023 Kahramanmaraş earthquakes. The lines and shading show the mean and standard deviation from the models listed in Table S1.

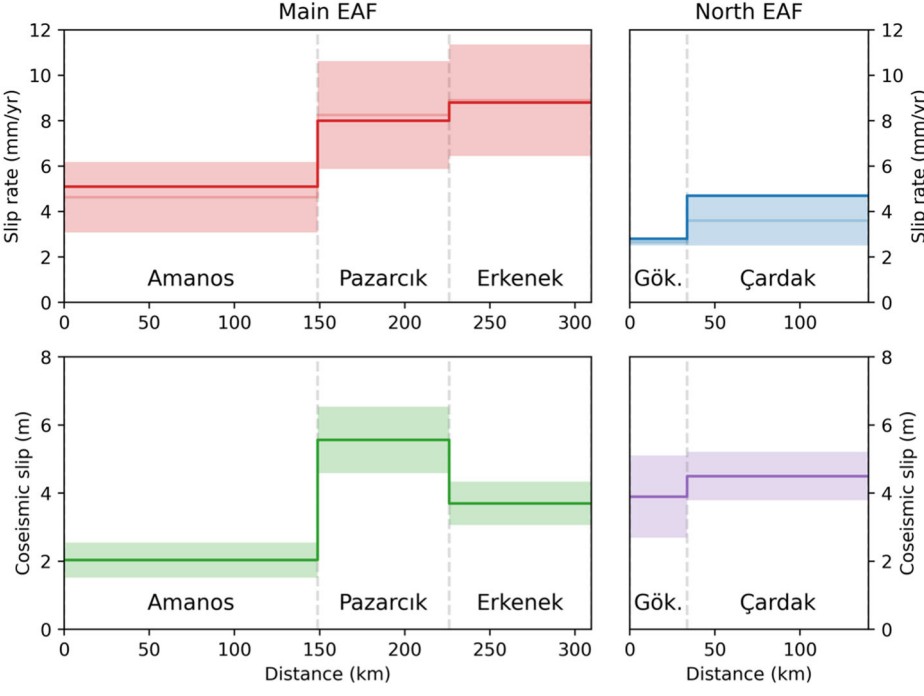

In general, we find good agreement amongst the analyzed finite fault models in terms of the relative and absolute amounts of coseismic slip during the 2023 Kahramanmaraş earthquakes. For the initial $M_w$ 7.8 mainshock, the Pazarcık segment hosted the largest amount of average slip of 5.6 ± 1.0 m. The slip along the Amanos and Erkenek segments are well-constrained at 2.0 ± 0.5 m and 3.7 ± 0.6 m, respectively. For the subsequent $M_w$ 7.7 event, the average slip is 4.7 ± 0.7 m on the Çardak segment, and 4.0 ± 1.2 m on the Göksun segment. The higher uncertainty on the Pazarcık segment may be due to tradeoffs with the adjacent Narli segment, where slip initiated during the $M_w$ 7.8 event[17]. Similarly, the discretization of the Göksun sub-orthogonal splay fault varies from model to model and possibly contributes to the larger associated uncertainty on the coseismic slip amplitude of 1.2 m. We additionally note that some variations in slip between different models may be affected by the time-windows of input data used, e.g., due to afterslip[60,61], although the contribution of postseismic deformation is likely to be small compared to the coseismic one.

## Interseismic strain accumulation

Estimation of the slip deficit on each fault segment that ruptured during the 2023 earthquake sequence requires knowledge of the long-term slip rates on the respective faults[62]. While the interseismic deformation along the EAF system has been the subject of numerous studies (Table S2), there remain large uncertainties on the long-term slip rates along the fault system (Fig. 3 and S2). On the main strand of the EAF, there is agreement that slip rates decrease toward the southwest, as the fault becomes less-aligned with relative plate motion between Anatolia and Arabia[35]. In the context of the 2023 earthquake doublet, perhaps the least well known is the slip rate on the GCFS, comprised of the Göksun, Çardak and Sürgü fault segments (Fig. 1). Ref. 29 suggest a slip rate of 2–3 mm/yr based on geomorphological offsets, but no independent estimates based on geodetic data are available to the best of our knowledge.

We use recently published secular velocities estimated from GNSS data[63] to refine estimates of the geodetic slip rates on fault segments that ruptured in February of 2023, with particular emphasis on the GCFS (see Methods for details). We extract profiles of interseismic velocities perpendicular to the respective fault traces, and invert them using a classic semi-infinite screw dislocation model in a homogeneous elastic half-space[62]. We employ a bootstrapping method to quantify the uncertainty associated with

the slip rate inversions (see Methods). The data and predictions of the best-fit models are shown in Figs. S3–S7. We also verified results of 2-D models using a 3-D model that accounts for variations in fault geometry and a full horizontal velocity field (see Methods and Fig. S8).

Our analysis shows that the Pazarcık segment, which hosted a 5.6 ± 1.0 m slip during the first mainshock, has a geodetic slip rate of 8.0 mm/yr (Fig. 2c and S3). To the northeast, the Erkenek segment, which hosted average coseismic slip of 3.7 ± 0.6 m, was accommodating a slightly higher geodetic slip rate of 8.8 mm/yr (Fig. 2b and S4). To the southwest, the Amanos segment, hosting average coseismic slip of 2.0 ± 0.5 m during the first mainshock, had a much lower geodetic slip rate of 5.1 mm/yr (Fig. 2d and S5). Along the rupture zone of the second mainshock, the Çardak segment, which hosted average coseismic slip of 4.7 ± 0.7 m, was moving at a rate of 4.7 mm/yr during the interseismic period (Fig. 2e and S6). The Göksun segment, which hosted average coseismic slip of 4.0 ± 1.2 m, was moving at a rate of 2.8 mm/yr (Fig. 2f and S7).

To provide further quantification on the uncertainty on the fault slip rates, we compare our estimated values to a collection of previously published geodetic and geologic slip rates (Fig. 3a, b and Table S2). Our slip rates for the Erkenek and Pazarcık segments, respectively 8.8 ± 0.3 mm/yr and 8.0 ± 0.5 mm/yr, are nearly identical to the mean of all previous estimates. The value of 5.1 ± 0.3 mm/yr for the Amanos segment is slightly larger but within errors with the 4.6 ± 1.5 mm/yr rate suggested by previous work[23–25,27,33–35,37,38,42,64] (Fig. 3a). While the 2.8 ± 0.5 mm/yr obtained for the Göksun segment is consistent with the 2–3 mm/yr rates inferred along the GCFS from field offsets[29], our estimate of 4.7 ± 0.3 mm/yr for the Çardak segment is larger by ~2 mm/yr (Fig. 3b). We note that if we examine geodetic and geologic slip rates separately along the EAF, there is a similar discrepancy of 2–3 mm/yr. It is likely that this is due to differences in the spatial aperture of different techniques. Geologic offsets are localized to a zone within meters to several kilometers of the fault trace, whereas geodetic data are measuring deformation on the scale of tens of kilometers or greater. Thus, geologic fault slip rates may be systematically lower as they may exclude inelastic strain accumulated off-fault[65] whereas geodetic slip rates may be over-estimating the actual slip rate in case of unmodeled contributions to secular velocity field[66] and/or transient deformation[67,68]. This is especially applicable to relatively immature strike-slip faults which tend to produce more deformation in a distributed manner[69,70], such as the EAF and

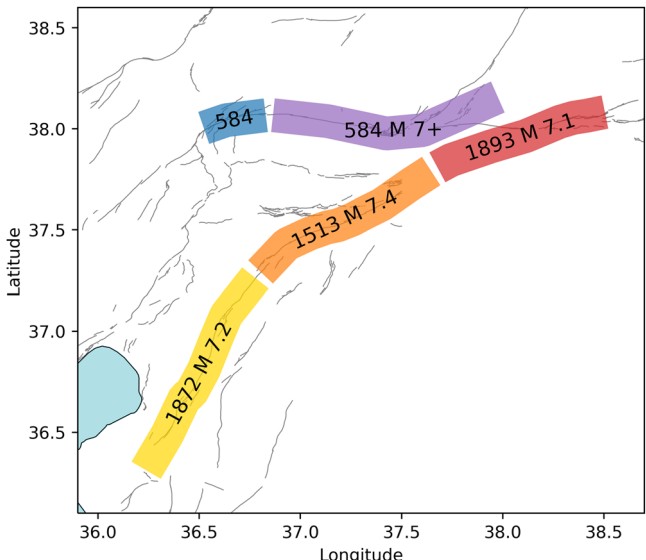

**Fig. 4 | Historic earthquakes along the East Anatolian Fault.** Dates and magnitudes of M > 7 earthquakes which preceded the 2023 Kahramanmaraş doublet on the Amanos (yellow), Pazarcık (orange), Erkenek (red), Çardak (purple), and Göksun (blue) segments of the East Anatolian Fault System.

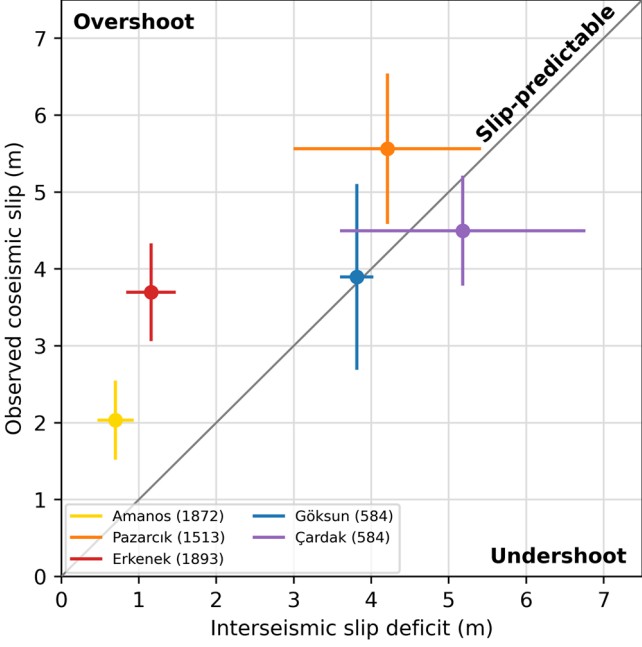

**Fig. 5 | A comparison of accumulated slip deficits since the penultimate earthquakes with coseismic slip during the 2023 Kahramanmaraş earthquake doublet.** The slip-predictable model suggests a one-to-one correspondence between coseismic slip and accumulated slip deficit, indicated by the dark gray line. Coseismic slip in excess of the slip deficit is termed overshoot, whereas undershoot refers to less coseismic slip than slip deficit. Color-coded circles denote various fault segments ruptured during the 2023 earthquake sequence. Error bars correspond to the standard deviation of our estimates of coseismic slip and slip deficit. Each segment is labeled with the date of the last major M > 7 earthquake.

GCFS with inferred total offsets of 11 km and ~20 km, respectively[29]. Another possibility is that the discrepancy is due to temporal variations in slip rate[71,72]. Therefore, for our analysis we take the average of all slip rate estimates (including those presented in this study) as the preferred values for the EAF and GCFS segments.

## Historic ruptures along the East Anatolian fault system

Historic records from a variety of sources cover over two millennia of major (M 7+) earthquakes in the vicinity of eastern Anatolia. Previous studies have established likely dates, rupture extents, and magnitudes based on descriptions of damage and the spatial distribution of reports[43–45,47,73,74]. Below, we outline the timing and nature of M 7+ events which have been inferred to directly precede the 2023 Kahramanmaraş earthquake doublet.

The most recent major earthquake on the EAF was a M 7.1 event in 1893 along the Erkenek segment, relatively well-documented in terms of rupture extent and shaking intensity[45]. Two large events occurred in the vicinity of the Amanos fault in the 19th century, a M 7.5 event in 1822 and M 7.2 in 1872[47,75]. While some previous studies have assumed these events occurred on the Amanos segment[45,47], others have suggested both occurred on other faults due to a lack of field evidence for the rupture on the Amanos. Other possible scenarios are a 1513 M 7.4 event was the preceding rupture prior to 2023[76] or that that the southern Amanos segment was ruptured by the 1872 M 7.2, while the 1513 M 7.4 occurred on the northern end[75]. There is strong evidence for the 1513 M 7.4 event along the Pazarcık segment from paleoseismic trenching, although inconclusive results at the easternmost trench site suggests that the full segment may not have ruptured[22]. Both trenches showed evidence for a very large M 7.8+ event in 1114[22,47], placing an upper bound on the timing of the last major earthquake along the Pazarcık segment.

Information regarding past ruptures along the GCFS is more limited. Paleoseismic trenching indicates that at least the Çardark and Sürgü segments experienced a large earthquake between ~1200–3200 years BCE[46]. It is likely that the GCFS is associated with an event in 584 or 587 CE that destroyed the settlement of Arabissus[46,47]. A M 6.7+ event has been inferred to have occurred in the vicinity of the GCFS in 1544[45]. However, the lack of paleoseismic evidence for this event along the Çardak segment in combination with the relatively low inferred magnitude led us to discount the 1544 event in terms of strain release along the GCFS.

We take the 1872 M 7.2 event on the Amanos segment, the 1513 M 7.4 on the Pazarcık segment, the 1893 M 7.1 on the Erkenek segment, and the 584 M 7+ on the GCFS to be the most likely penultimate major earthquakes along the main fault segments ruptured during the 2023 $M_w$ 7.8–7.7 earthquake sequence (Fig. 4). Unless otherwise noted, the subsequent discussion assumes this historic rupture scenario. Several alternative scenarios are presented in the Supplementary Discussion (Figs. S9–12).

## Results and discussion
### Comparing slip deficit with coseismic slip
The slip-predictable model for earthquake recurrence suggests that when failure occurs along a fault, the slip should equal the slip deficit accumulated since the preceding event[1,2]. A less strict interpretation of slip predictability would be a correlation between the earthquake magnitude and recurrence interval[10,11,15]. If found to be applicable to a given fault system, the implication for hazard assessment would be the capacity to forecast the magnitude of a future earthquake given the fault slip rate and date of last rupture. To estimate the accumulated slip deficit along the different segments of the EAF, we multiply the fault slip rate by the time elapsed between the preceding M 7+ earthquake (Fig. 4) and the 2023 Kahramanmaraş doublet. We propagate the standard deviation on the fault slip rate estimates (Fig. 3) to quantify the uncertainty on the slip deficit for a particular historical earthquake.

Comparing the estimated slip deficit and the average coseismic slip from finite fault inversions (Fig. 5) reveals several salient features. First, there is a clear positive correlation between the magnitude of the accumulated slip deficit and coseismic slip, suggesting that the latter depends on the accumulated elastic energy since the last rupture, consistent with loose slip-predictability. Second, there is an apparent distinction in the behavior between the EAF and GCFS. On the GCFS (the Çardak and Göksun segments), the accumulated slip deficit and coseismic slip are essentially equal within the data uncertainties, suggesting that the rupture along these segments was slip-predictable to the first order. On the EAF (the Erkenek,

**Table 1 | Slip, rates, and dates of earthquakes along the 2023 Kahramanmaraş ruptures**

| Fault segment | Coseismic slip (m) | Slip rate (mm/yr) | Penultimate event date | Postdicted date |
|---|---|---|---|---|
| Pazarcık | 5.6 ± 1.0 | 8.0 ± 0.5 | 1513 | 1348 ± 780 |
| Erkenek | 3.7 ± 0.6 | 8.8 ± 0.3 | 1893 | 1607 ± 484 |
| Amanos | 2.0 ± 0.5 | 5.1 ± 0.3 | 1872 | 1588 ± 500 |
| Çardak | 4.7 ± 0.7 | 4.7 ± 0.3 | 584 | 717 ± 1600 |
| Göksun | 4.0 ± 1.2 | 2.8 ± 0.5 | 584 | 485 ± 1120 |

Postdicted dates are obtained by dividing the observed coseismic slip by the fault slip rate.

Pazarcık, and Amanos segments), we find that coseismic slip is generally larger than the accumulated slip deficit, consistent with estimates from earlier studies[77]. Interestingly, the Pazarcık segment, which had larger coseismic slip, more closely follows the slip-predictable model, while the Erkenek and Amanos segments (more distal from the epicenter) produced less slip but appear to exhibit overshoot (slip in excess of the accumulated slip deficit since the penultimate event). Equivalently, we can postdict the dates of preceding earthquakes by dividing the observed coseismic slip by the inferred slip rates and the respective time intervals from the year 2023. For the GCFS, we find that the postdicted dates are within ± 150 years of the M 7 + 584 earthquake, although the uncertainties are large (Table 1). For the EAF, the postdicted earthquake dates are systematically earlier than the actual dates, implying that the $M_w$ 7.8 event released extra strain equivalent to several hundred years of interseismic strain accumulation with respect to the slip-predictable model.

The observation of slip overshoot on the EAF indicates that the preceding major earthquakes that were limited to individual fault segments, namely the 1893 M 7.1 Erkenek, 1513 M 7.4 Pazarcık, and 1872 M 7.2 Amanos events (Fig. 4), left residual stresses on the faults which were released, at least in part, during the 2023 event, in addition to stresses accumulated in the intervening period. This may be due to the preceding earthquakes not rupturing the full extent of each fault segment[22,75], incompletely releasing the accumulated slip deficit during a given rupture, or a combination of both. On the GCFS, our results indicate that the $M_w$ 7.7 event was approximately slip-predictable (Fig. 5). Overall, fault segments that accumulated a higher slip deficit appear to be in better agreement with the slip-predictable model (Fig. 5), while those with a smaller slip deficit produced overshoot, possibly enabled by the cascading nature of the $M_w$ 7.8 event. The release of residual stresses by the $M_w$ 7.8 earthquake demonstrates how rare cascading events can pose a much larger seismic hazard than suggested by paleoseismic data spanning a limited number of past events.

## Discussion of uncertainties

Given uncertainties in paleoseismic records, we evaluated several alternative earthquake histories (see Supplementary Discussion). If the Pazarcık segment last ruptured in 1795, its slip deficit would decrease and the overshoot effect would be enhanced (Fig. S9). If the Amanos segment ruptured in 1513, rather than in the 19th century, it would become approximately slip-predictable (Figs. S10 and S12). The only scenario in which segment ruptures do not exhibit slip overshoot or slip-predictability is if the M 7.8+ earthquake in 1114 was the penultimate event on the Amanos or Pazarcık segments (Figs. S11 and S12). However, data from paleoseismic trenching indicates that the Pazarcık segment ruptured, at least partially, in 1513[22].

As noted above, the magnitude of the estimated mean coseismic slip depends on the assumed locking depth of each fault segment; decreases in locking depth increase the average slip-per-segment, and vice versa. So, opting to use the 90% seismicity cutoff depth or geodetically-derived locking depths (Fig. S1) results in an average increase in slip of 26% and tends to promote slip overshoot (Fig. 5). Conversely, the 99% cutoff depth reduces the average coseismic slip by 27% and promotes slip undershoot.

We also consider the possibility of increasing fault slip rates over time, which might reconcile discrepancies between geologic and geodetic slip rates (Fig. S2). We note that acceleration of interseismic deformation over the observed period (up to ~1400 yrs) preceding the 2023 ruptures would be similar in duration to intervals of inferred changes in slip rate along other fault systems[71]. However, acceleration over this interval would not impact our slip deficit estimates (see Supplementary Discussion). If time-dependent patterns of relative seismic activity and quiescence along the EAF system are interpreted as arising from periods of high and low slip rates, then historical earthquake catalogs suggest accelerated slip during ~500–1100, then followed by deceleration after ~1100[78]. This conflicts with the observed higher geodetic (i.e., present-day) slip rates and lower geologic slip rates (Fig. S2); thus, we conclude that temporal variations in slip rates, if any, do not affect our conclusions.

## Slip overshoot, rupture length, and cascading effects during the 2023 Kahramanmaraş earthquake doublet

Our results indicate that on average coseismic slip on the Amanos, Pazarcık, and Erkenek segments exceeded the slip deficit accumulated since the preceding M > 7 earthquakes by 1−2 m (i.e., exceeded the prediction of the slip-predictable model; Fig. 5). This implies that these penultimate events did not result in complete stress drops on the faults at their respective times of rupture. We suggest that the robust dynamics of the 2023 $M_w$ 7.8 event promoted rupture propagation over large distances, and facilitated the release of at least some residual stresses left behind by the penultimate events. In particular, the $M_w$ 7.8 event ruptured across several potential barriers, such as the Narli and Erkenek fault junctions, and involved unfavorable backward propagation along the Pazarcık and Amanos segments due to dynamic unclamping[17,79].

The ability of the rupture to break through multiple structural barriers indicates that it was highly energetic, possibly due to activation of enhanced dynamic weakening[80,81]. This may explain a higher stress drop compared to stress drops due to smaller (segment-bounded) ruptures that may be less efficient in producing strong dynamic weakening. Note that the most distal fault segments with respect to the hypocenter—the Amanos and Erkenek— had the largest relative slip overshoot with respect to the slip deficit accumulated since the preceding events (Fig. 5). This is consistent with a progressive dynamic weakening of a propagating rupture with time. It was also suggested that spatial heterogeneities in frictional properties may result in alternation of segmented and "wholesale" rupture scenarios[82–84]. Wholescale fault ruptures can erase the slip deficit accumulated in fault sections between the nominal asperities[85,86].

The $M_w$ 7.7 event had a larger stress drop and a smaller rupture length compared to the $M_w$ 7.8 event, and to the first order is consistent with the slip-predictable model (Fig. 5). In part, the inferred differences in release of the accumulated elastic strain energy (slip-predictable versus overshoot) may be due to differences in fault maturity between the EAF and the Sürgü-Çardak-Göksun strand. Namely, the more mature (higher offset) EAF[29] is more likely to produce long multi-segment ruptures. We also note that the $M_w$ 7.7 event was brought closer to failure and likely triggered in some way by $M_w$ 7.8 event[17,87]. While the aftershock statistics is reasonably well known, the occurrence of similar-size earthquakes on nearby faults ("earthquake doublets") is less well understood, which poses challenges for accurately forecasting seismic hazards[88].

Seismic hazard along the EAF was well-recognized prior to the 2023 Kahramanmaraş earthquake doublet[89]. Our analysis confirms that slip along each of the main structural fault segments that participated in the $M_w$ 7.8 and $M_w$ 7.7 ruptures at minimum coincided with the slip deficit which had accumulated since the preceding major earthquakes. However, the $M_w$ 7.8 relieved strain greatly in-excess of the corresponding slip deficit, emphasizing that our incomplete knowledge of fault ruptures over multiple seismic cycles challenges our ability to estimate the maximum possible magnitude of future earthquakes on a given fault. Our results highlight the importance of characterizing earthquake rupture propagation and cascading effects not only for determining the overall rupture dynamics and strain release during

major earthquakes, but also understanding the long-term slip behavior over many seismic cycles.

## Conclusions

In this study, we investigate the relationship between interseismic strain accumulation and occurrence of major (M 7+) earthquakes along the segments of the East Anatolian fault (EAF) that ruptured in the 2023 $M_W$ 7.8–7.7 Kahramanmaraş earthquake doublet. Given information on the fault's behavior throughout one major earthquake cycle, including the dates of past ruptures, rates of interseismic strain accumulation, and detailed data documenting the 2023 coseismic process, we evaluate the applicability of the slip-predictable model of earthquake recurrence. We make estimates of the slip deficit which had accumulated since the preceding M 7+ earthquakes along the EAF and coseismic slip, along with corresponding uncertainties. Across a range of possible historic rupture scenarios based on historic records and paleoseismic studies, we find that coseismic slip in 2023 was slip-predictable as a lower bound, but in places exceeded the accumulated slip deficit. These findings highlight the challenges of forecasting the maximum earthquake magnitudes even with knowledge of past events, as well as reciprocal complexity between rupture evolution and long-term seismic cycle behavior.

## Methods

### Locking depth estimation

Previous work has produced various estimates for locking depth along the EAF system. Inversions of interseismic surface velocities from InSAR have suggested locking depth as shallow as several kilometers along the northeastern portion of the fault, and as deep as 26 km along the Amanos segment[23]. Analyses of earthquake focal depths have indicated the seismogenic depth range extends to 18–20 km along the EAF[57]. To independently constrain the locking depths, we use a new catalog including both background seismicity from 2018–2023 and aftershocks of the 2023 sequence to more precisely constrain the locking depth of each fault segment that participated in the 2023 Kahramanmaraş doublet. We select events located laterally within 20 km of each fault segment trace and compute the 90%, 95%, and 99% catalog cutoff depths[90].

We find that the cutoff depth is quite sensitive and the difference between the 90% and 99% thresholds is typically over 10 km, ranging from 14.4–17.2 km to 23.9–28.0 km, respectively (Fig. S1). However, the estimates made using a 95% cutoff align well with the bottom edge of coseismic slip as resolved by finite slip models (Fig. S1). Given the general agreement between these two datasets, we adopt the 95% cutoff depths for our preferred locking depths. These values tend to be somewhat larger than those obtained from 2D inversions of GNSS data (Figs. S4–S8). Since the GNSS station spacing is of similar order to the estimated locking depths (10–30 km), the latter may not be well resolved by the data, and we prefer the locking depths constrained by seismicity and models of coseismic deformation.

### Modeling of interseismic deformation

We use recently published secular GNSS velocities[63] to estimate geodetic slip rates on faults that ruptured during the 2023 earthquake sequence. The velocity data were inverted using a buried dislocation model, wherein we assume a homogeneous elastic medium and use the fault-parallel component of the velocity field[62,91]. We identified five quasi-linear fault segments (Fig. 1) that are long enough to be adequately approximated by a 2-D antiplane strain model (i.e., the along-strike dimension several times larger than the locking depth). For each segment, we defined a fault-perpendicular profile extending for at least 100 km to ensure that the asymptotic nature of the fault-parallel velocities is adequately captured, and extracted the GNSS velocity data along the profile. Estimating slip rates due to closely-spaced sub-parallel strike-slip faults is challenging due to strong trade-offs in the model parameters[92]. For this reason, we take several precautions in estimating the slip rates along the Çardak and Pazarcık segments (Fig. 1). We note that the profile for the Çardak segment is truncated prior to its intersection with the Pazarcık segment in order to avoid fitting the larger signal

associated with the EAF—we further assess the validity of this choice below. We then solved for the best-fit slip rates and locking depths for each fault segment using a grid search algorithm.

To assess the uncertainty of the slip rates and locking depths, we utilized a bootstrap approach. During each iteration, we perturb the original profile by randomly re-sampling each data point within a range described by the corresponding model misfit and then invert the perturbed data. We repeat the bootstrap procedure 100 times and compute the standard error of the resulting bootstrap solutions. We then estimate the uncertainty for a 90% confidence level by multiplying the bootstrap standard error by the corresponding $t$-value of 1.66 (for 100 samples). Results of the inversions are shown in Fig. 2 and S4–8.

To evaluate the effects of 3-D fault geometry and potential trade-offs between slip rates on closely spaced faults (i.e., the Çardak and Pazarcık segments), we also construct a 3-D fault model using rectangular dislocations in a homogeneous elastic half-space[93] (Fig. S8). The model assumes locking depths suggested by the 2-D inverse models. We account for the variable dip angle of the Çardak segment, as suggested by the aftershock hypocenters. For simplicity, we allow no variations in slip rate along the Çardak segment and the EAF to the east of the $M_w$ 7.8 epicenter (green and orange lines in Fig. S8, respectively). We also account for contributions of the NAF and the Dead Sea Fault (DSF) to the GNSS velocity field by prescribing secular slip rates of 22 and 2.5 mm/y, respectively, and the Central Anatolian Fault Zone (CAFZ), on which the slip rate is found as part of the solution. We then invert the full horizontal velocity field for the best-fit slip rates on the EAF, CAFZ, Amanos, and Çardak fault segments using least squares. Results are shown in Fig. S8. Despite a relative simplicity of the model, it is able to explain the overall velocity pattern reasonably well. It also reveals a good agreement with slip rate on the Çardak segment suggested by a 2-D model (Fig. S6). Our 3-D model suggests somewhat higher slip rates on the EAF and Amanos segments compared to the 2-D results, however we note that the former over-predicts the observed fault-parallel (strike-slip) velocity components (Fig. S8), possibly due to an oversimplified geometry of the regional fault network. The 2-D models that consider the fault-parallel velocity component only (Fig. 2) are likely a better representation of the geodetic fault slip rate.

## Data availability

Aftershock locations are available from KOERI. Interseismic GNSS data can be accessed via Aperta Türkiye Open Archive (https://aperta.ulakbim.gov. tr/record/252408#.ZF35yXZByUl). Finite fault models and fault slip rates used in this study are available from the original publications (see Tables S1 & S2).

## Code availability

Codes for geodetic inversions and analysis of finite slip models used in this study are available upon request.

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

## Acknowledgements
Aftershock locations were obtained from KOERI. Maps and graphs were generated using GMT and MATLAB. The authors wish to thank Doğan Şeber, Derya Gürer, and Joe Aslin for their editorial feedback, as well as several anonymous reviewers whose thoughtful comments helped improve this study. We acknowledge support from NASA (80NSSC22K0506 to Y.F.) and NSF (GRFP to E.V.).

## Author contributions
E.V. analyzed the slip models and seismicity and performed inversions of interseismic velocities using a 3-D fault model. F.B. performed 2-D inter-seismic modeling of GNSS data. E.V., Y.F., F.B., A.G., S.Y., and C.Y. discussed the results and contributed to the writing of the manuscript.

## Competing interests
The authors declare no competing interests.
