## [Transparent Peer Review file · Communications Earth & Environment]

The 2023 M_w 7.8-7.7 Kahramanmaraş earthquakes were loosely slip-predictable

Corresponding Author: Mr Ellis Vavra

Version 0:

Decision Letter:

Dear Dr Bulut,

Your manuscript titled "The 2023 MW 7.8-7.6 Turkey Earthquakes follow the slip-predictable model" has now been seen by 3 reviewers, whose comments are appended below. You will see that they find your work of some potential interest. However, they have raised quite substantial concerns that must be addressed. In light of these comments, we cannot accept the manuscript for publication, but would be interested in considering a revised version that fully addresses these substantial concerns.

We hope you will find the reviewers' comments useful as you decide how to proceed. Should additional work allow you to address these criticisms, we would be happy to look at a substantially revised manuscript. If you choose to take up this option, please either highlight all changes in the manuscript text file, or provide a detailed list of the changes to the manuscript with your responses to the reviewers.

In particular, we would need to see that the revised manuscript meets the following editorial thresholds:

- Define a research question and place your findings in the light of existing data and published literature. In this context, it is also important to provide magnitudes of historical earthquakes with adequate references.

- Quantify uncertainty estimates or evidence for robustness of the presented model and document the steps in the analyses done. This approach will ensure your data reproduced and your methodology can be deployed elsewhere.

- In terms of increasing the potential for scientific impact beyond the recent earthquakes, I encourage you to first document your numerical simulation approach and then either provide additional numerical simulations (or references to published work on fault systems with slip predictability). This addition will allow to expand on the possibility that the EAF and the NAF could be natural examples of the most simple bounding case for slip predictability and increase the impact of your contribution.

If the revision process takes significantly longer than three months, we will be happy to reconsider your paper at a later date, as long as nothing similar has been accepted for publication at Communications Earth & Environment or published elsewhere in the meantime.

We understand that due to the current global situation, the time required for revision may be longer than usual. We would appreciate it if you could keep us informed about an estimated timescale for resubmission, to facilitate our planning. Of course, if you are unable to estimate, we are happy to accommodate necessary extensions nevertheless.

We are committed to providing a fair and constructive peer-review process. Please do not hesitate to contact us if you wish

to discuss the revision in more detail.

Please use the following link to submit your revised manuscript, point-by-point response to the reviewers' comments with a list of your changes to the manuscript text (which should be in a separate document to any cover letter), a tracked-changes version of the manuscript (as a PDF file) and any completed checklist:

Link Redacted

Please do not hesitate to contact us if you have any questions or would like to discuss the required revisions further. Thank you for the opportunity to review your work.

Best regards,

Derya Gurer, PhD
Editorial Board Member
Communications Earth & Environment
orcid.org/0000-0001-5884-9160

Joe Aslin
Senior Editor
Communications Earth & Environment

EDITORIAL POLICIES AND FORMAT

If you decide to resubmit your paper, please ensure that your manuscript complies with our editorial policies and complete and upload the checklist below as a Related Manuscript file type with the revised article:

Editorial Policy Policy requirements
(Download the link to your computer as a PDF.)

For your information, you can find some guidance regarding format requirements summarized on the following checklist: (<https://www.nature.com/documents/commsj-phys-style-formatting-checklist-article.pdf>) and formatting guide (<https://www.nature.com/documents/commsj-phys-style-formatting-guide-accept.pdf>).

REVIEWER COMMENTS:

Reviewer #1 (Remarks to the Author):

This manuscript uses regional GPS observations to invert for the slip model of the 2023 Türkiye earthquake sequence. With the slip models, it also presents Coulomb stress changes and compares the inverted amount of slip to the historical earthquake record, supporting a hypothesis of slip-predictable events in the region. Each of these results is derived from well-established methods and appropriate observational constraints. The first two, the details of event displacement and consequent regional static stress changes, are consistent with other studies of the earthquakes. The latter is a more surprising result since few, if any, major faults have been shown to be slip-predictable.

The first two major results, therefore, are consistent with and incremental to closely related studies. The approach has been used on hundreds of moderate to large earthquakes globally. As a result, a study like this would normally be published in topical or regional journals. The primary justification for publication in Nature Earth and Environment Communications is the rapid reduction of data combined with the societal impact of the earthquake sequence. Because these events had disproportionate impacts to human communities, they are of particular and timely scientific interest. The contributions of the slip model and stress change results to general earthquake mechanics, seismology, or tectonics are incremental.

The third major result, of slip-predictability, may have broader scientific impacts. Because fault systems are complex and highly nonlinear, slip-predictable seismicity over multiple earthquake cycles has seldom been observed in the seismic record. Instead, most theoretical models and observational constraints indicate that the amount of elastic slip potential stored on a fault system since the previous rupture should not be used to determine the likely magnitude or timing of the next event. However, the East (EAF) and North Anatolian Fault (NAF) may differ from "typical" fault systems by having simpler behavior. For example, the NAF ruptured in a nearly-linear temporal and spatial sequence in the 20th century, providing the most famous example of earthquake triggering in seismology. If this structured behavior is the consequence of structural simplicity or a more linearized frictional law for the system, one might expect rupture behavior closer to the slip-predictable end member. This possibility is more scientifically novel, and perhaps worthy of more development and emphasis in the manuscript.

I would therefore suggest two editorial options for the manuscript. If the editor would like to get a paper out about the Türkiye events quickly because of their non-scientific impacts, the paper could be published in its current form with minor grammatical and style revisions. If, on the other hand, the editor would prefer to see a manuscript with more potential for scientific impacts, the authors should be encouraged to incorporate more numerical simulation (or citations from prior work on fault systems with slip predictability) to expand on the possibility that the EAF and the NAF could be natural examples of the most simple bounding case for slip-predictability.

Reviewer #2 (Remarks to the Author):

This manuscript presents the integration of various analysis to study the February 2023 earthquake doublet. Time series analysis of data from a network of continuous GPS stations that registered the earthquake allowed the estimation of coseismic displacements at the GPS sites for each of the earthquakes. These displacements were then inverted to obtain a fault slip model for which the various ruptured segments were defined by the aftershock distribution. The preferred model was validated by comparing surface displacements along the rupture measured from orthophotos with model predictions at the same locations. For each one of the segments that intervened in the earthquakes, the authors modeled the interseismic slip rate and locking depths using a published large scale velocity field. The slip rate estimates and the average coseismic slip allowed to bracket the time required to accumulate the appropriate slip deficit, which was in good agreement with the dates of the last historical earthquakes known for each segment. This provided a second validation of their slip distribution model. This result suggests that the doublet was consistent with a slip-predictable model for this mature fault system, rather than a characteristic earthquake or time predictable model. The coseismic slip model was also used to estimate the Coulomb stress changes imposed by each of the earthquakes on nearby faults. The second main shock and the aftershocks of both earthquakes were consistent with faults with increased stress. Moreover, the combination of the two earthquakes produced large Coulomb stress changes on several faults, which likely are now closer to failure.

The manuscript is very well written and is a pleasure to read through. I found the results very compelling. If not for a few details, the paper could be ready for publication as is. However, I noted a few places where the discussion could be refined and a few figures could be improved slightly.

Find my comments in the attached file.

Apart from these minor comments, please accept my congratulations on a very interesting paper.

Reviewer #3 (Remarks to the Author):

The authors analyze the 2023 Kahramanmaraş earthquake sequence in context of the pre-earthquake interseismic slip rates and the stress effects of these earthquakes on other faults in the region. Their coseismic analyses, while similar to many that have been posted or published previously, appear robust. Their static stress change calculations also essentially reproduce what has been done previously, but the authors do extend their analysis to other potentially loaded faults in the region. The authors conclude based on a comparison of interseismic fault slip rates and coseismic slip magnitudes that large earthquakes in this region slip the entire accumulated slip deficit, a result that if true would be an important contribution to understanding earthquake processes. However, the manuscript has several areas that could use substantial improvement. The referencing is very light and the authors do not define a key scientific question that they are trying to answer. I am not convinced I could reproduce the authors' results from their descriptions, and I have concerns about the validity of the interseismic modeling. Finally, all of the figures could be improved. I provide specific comments and suggestions below, and I would be happy to look at a revised version of this manuscript.

Major Comments

In my opinion, the authors are missing a main scientific question in the Introduction. They include a description of the tectonics, the recent earthquake sequence, and their analytical approaches/results. But I think their paper would be strengthened by setting up a research question that their work answers. The manuscript is also missing some references. The authors are not the first to look at the coseismic displacements for these events, generate slip models for these earthquakes, interpret the main event as starting on a smaller, non-EAF fault, or calculate Coulomb stress changes. I highly recommend that the authors look at some of the other studies that have been done already and examine their results in light of those other analyses.

The authors make several assertions about their slip inversion results that may be true but appear to be unsupported by evidence. Specifically, they call their model "relatively low resolution" (l. 121), but do not define what this means or how they determine its resolution. They say the "slip distribution is necessarily smoothed" (l. 121) but provide no reasoning for applying smoothing constraints or how much or by what means the model is smoothed. Finally, they claim that "the spatially averaged slip...is well constrained" (ll. 122-123), but provide no uncertainty estimates or evidence for robustness. Even if these show up later in the Methods section, it is important for the authors to provide some quantitative support to these statements here.

I have a couple of concerns about the authors' interseismic and coseismic inversion approaches. I am not sure the authors

provide enough information in all cases that I could reproduce their results, and I am very familiar with these types of inversions. For example, the assumptions underlying the “buried dislocation model” (l. 490) should be described, perhaps with a formula. How long is a fault before it is “adequately approximated by a 2-D antiplane strain model” (ll. 491-492)? The profile explanation (ll. 492-493) does not explain the selected width of velocities around the profile or whether profiles intersect multiple faults. Finally, the description of uncertainty analysis is too full of jargon and does not explain the uncertainty estimation approach well (ll. 494-497). Similar gaps are in the coseismic modeling description. Another concern I have about the interseismic modeling is whether it is done correctly. In Supp. Fig. 4, the profile crosses the EAF and the Çardak Fault to the north, yet I see no jump in velocities across the Çardak Fault as shown in Supp. Fig. 7. I am not sure this is completely explained by the projection of velocities onto the fault. Then, in Supp. Fig. 7, I do not understand why the authors cut the profile off at the Pazarçik segment of the EAF. In addition, they show a 9 mm/yr difference across the Pazarçik segment, half of which is completely absent here. I think the authors need to make sure this analysis is done correctly.

The slip-predictable earthquake model states that each earthquake releases all of the accumulated slip deficit. But that also assumes that we know what the accumulated slip deficit is (not just the slip deficit rate). I suspect that M 7.5+ earthquakes in strike-slip systems may locally release most of the accumulated slip deficit and essentially reset the clock. However, I am less confident that M 7.0 (e.g., l. 252) or smaller events do the same. They will leave slip deficit on the fault. Related to this, the authors do not provide magnitudes for all of the historical events they discuss, which to me is a major omission.

Several of the figures probably need substantial improvement before publication. Fig 1a and 1b appear to be time series, but I find these almost unreadable. Arrows for displacements should not go off the map, and faults should be more distinguishable from displacement observations in Fig 1c. Why does Fig. 1d need to show the displacements at all? Lastly for Fig. 1, the authors should specifically refer to these subpanels in the manuscript. For Fig. 2, velocities derived from GPS are not “slip rate,” so these labels should be changed. “Latitude” is spelled incorrectly. This figure is not referred to in the manuscript until the Methods, so it should come later. The basemap for Fig. 3 is too low resolution. The authors should include negative values for Coulomb stress changes.

Minor Comments

The names of the major faults (EAF, NAF) are well established and known by the geological community. The segments and other faults are not as well known and therefore are not particularly meaningful to a general audience. If the authors choose to use these segment names, I think they should define the segment locations in the Introduction with a companion map. I also think it would be helpful to indicate whether these are broadly accepted segment labels or if the authors are naming the segments here for the first time.

l. 26, l. 44, l. 56: The term “doublet” carries a lot of implicit meaning, and different people use the term differently. I suggest defining the specific usage here or using another term. It becomes particularly confusing when talking about the doublet being “the largest seismic event in the region” (l. 57). Are the authors referring to combined seismic moment release? The initial, larger earthquake?

ll. 73ff: Aftershocks are not mentioned at all in the Introduction, nor are they introduced at the beginning of this section. I think the authors should motivate why aftershocks are relevant to discuss here instead of leaving that unsaid.

I am not familiar with what information should be in the Methods section at the end of the manuscript and what material should be in the Supplementary files. My instinct is that if a method is critical for an argument in the paper, it should be in the main body, not a supplement. Then, the authors should refer to the Methods section explicitly in the main body of their paper (e.g., when talking about the GPS displacements in ll. 96-97 or the slip model inversions in ll. 110-111). Otherwise, I feel like the manuscript is missing a lot of information that is probably included elsewhere.

I had difficulty following the figures because the numbering is out of order in the manuscript. I also found no reference to Figure 2. Please correct this.

It is unclear to me why the authors determine the background stress field (ll. 165-168). I do not see how they use these results in their analysis. Calculating stress changes generated by the earthquakes on faults with known geometry usually precludes the need to calculate background stresses (which is generally done to determine the optimal fault orientation when fault geometries are unknown). Also, I think it might be worthwhile for the authors to show faults with negative Coulomb stress changes, because earthquakes on these structures might be delayed.

The “Discussion” section in ll. 201-226 is not much of a discussion at all. I think if the authors expanded this section into a fuller comparison between their analyses and other analyses, this could be a valuable element of the discussion. But as it stands, this section reads more like the end of a Results section than a useful discussion of the implications of the analysis.

Detailed Comments

l. 27: I do not feel it is appropriate to combine the length of the two fault ruptures, since they are clearly different geological structures that had distinct earthquakes

l. 41: The “entire southern section” is too ambiguous a description. Please define the extent of the fault that ruptured and/or refer to the region on a figure.

II. 51-55: I think this sentence is too long and should be broken into 2 for clarity. I also think that the Hellenic subduction zone, while relevant to regional tectonics, might not be particularly relevant to these earthquakes near the EAF.

I. 56: What region are the authors referring to specifically for the frequency of M7+ earthquakes? Turkey as a country? The Anatolian Plate boundaries?

I. 62: I assume the continuous GPS data are daily positions, but this should be stated explicitly

I. 66: I would disagree with the characterization of interseismic deformation measured with GPS as "long term"

I. 84: Start a new paragraph with "The aftershock distribution..."

I. 96: The term "processed" does not capture the information I am most interested in from the GPS. I recommend the authors explicitly indicate that they determined coseismic displacements.

I. 108: Please define "most reliable." Are the authors referring to the sampling rate? The data distribution? The ability to model coseismic deformation from the dataset?

II. 189f: I found it a little unusual to jump between fractions of MPa and multiple KPa for the Coulomb stress changes. I would prefer to stick with one unit and describe all of the results with that unit.

I. 221: Reference for dynamic stress changes being larger than static?

II. 222-223: What is the relevance of a lack of foreshock activity to the Coulomb stress change results?

I. 224: The word "governed" is too non-specific

Communications Earth & Environment is committed to improving transparency in authorship. As part of our efforts in this direction, we are now requesting that all authors identified as 'corresponding author' create and link their Open Researcher and Contributor Identifier (ORCID) with their account on the Manuscript Tracking System prior to acceptance. ORCID helps the scientific community achieve unambiguous attribution of all scholarly contributions. You can create and link your ORCID from the home page of the Manuscript Tracking System by clicking on 'Modify my Springer Nature account' and following the instructions in the link below. Please also inform all co-authors that they can add their ORCIDs to their accounts and that they must do so prior to acceptance.

If you experience problems in linking your ORCID, please contact the Platform Support Helpdesk.

Version 1:

Decision Letter:

Dear Mr Vavra,

Your manuscript titled "Were the 2023 Mw 7.8-7.7 Turkey Earthquakes Slip-Predictable?" has now been seen by our reviewers, whose comments appear below. In light of their advice we are delighted to say that we are happy, in principle, to publish a suitably revised version in Communications Earth & Environment.

We therefore invite you to revise your paper one last time to address the remaining concerns of our reviewers. At the same time we ask that you edit your manuscript to comply with our format requirements and to maximise the accessibility and therefore the impact of your work.

EDITORIAL REQUESTS:

****Please take care to match our formatting and policy requirements. We will check revised manuscript and return manuscripts that do not comply. Such requests will lead to delays. ****

SUBMISSION INFORMATION:

OPEN ACCESS:

Communications Earth & Environment is a fully open access journal. Articles are made freely accessible on publication. For further information about article processing charges, open access funding, and advice and support from Nature Research, please visit <https://www.nature.com/commsenv/open-access>

Link Redacted

Best regards,

Derya Gürer, PhD
Editorial Board Member
Communications Earth & Environment
orcid.org/0000-0001-5884-9160

Joe Aslin
Deputy Editor,
Communications Earth & Environment
<https://www.nature.com/commsenv/>
Twitter: @CommsEarth

REVIEWERS' COMMENTS:

Reviewer #1 (Remarks to the Author):

This manuscript compares the destructive 2023 Türkiye earthquake doublet to conceptual physical models of earthquake energy budgets, specifically slip-predictable and time-predictable end members. This particular event doublet is a rare candidate for such comparison because the characteristics of the two events are well-constrained by observational data and past events are also relatively well-known in the region through a long historical record. The question of slip- or time-predictability is fundamental to understanding earthquake nucleation and energy budgets in earthquake physics and to evaluating earthquake risk and hazard to human communities. Therefore, the topic has profound broad interest and impact.

The authors do an unusually good job of explaining limiting models both succinctly and accurately, and demonstrating how this particular set of events serves as a reasonable test. Because large earthquakes are rare, and those with both sufficient historical and observational data even rarer, models of limiting reoccurrence are predominantly theoretical. Each new well-observed event therefore provides critical data for our broader understanding of the underlying physics.

It appears that this manuscript has already gone through at least one round of reviews and revisions, so overall it is nearly ready for publication. I can mostly contribute by drawing the authors' attention to a few papers that offer an explanation for the variability of moment magnitudes and their relationship to stored elastic energy or apparent slip overshoot. Specifically, Feldl and Bilham (2006) and a few others have pointed out that larger magnitude events may actually release mechanical potential energy (elastic or elastio-plastic) from larger volumes than smaller events, where the fault length scaling varies with L , the moment with L^2 , but the "reservoir" of stored energy with L^3 . It is therefore not entirely straightforward to convert the accumulated tectonic strain into slip potential because the former value is sensitive to the length of the baseline over which that strain is measured but larger events might actually discharge potential from a longer baseline or, actually, a larger volume. This effect is expected to produce the observations that the authors note: that larger events might look like they are overshooting the stored strain because that storage is not measured over a long enough baseline. This effect appears to be more important in continental settings, where strain is stored on faults and in volumes, where faults are less mature, and where there is a lot of kinematic complexity. All of these conditions apply to the East Anatolian Fault, so this study is a nice example of both the relevance of slip-predictability and the potential for nonlinear modulation of that slip-predictability. In my opinion, this added nuance could be enhanced with just a sentence or two and a citation or two, but it would increase the impact of the work.

One other effect that is the target of a lot of recent attention is that of synchronization or entrainment of triggered cascades, leading to effects like supercycles in synthetic catalogs (Field et al., 2017; Field et al., 2022; Milner et al., 2022; Brodsky and van der Elst, 2014; Van der Elst and Brodsky, 2010). This probably also warrants a sentence or two, since the event is a doublet consisting of events expected to have very different characteristic timescales. Even if multicycle temporal alignment is considered unlikely (i.e. this set of faults does not have a previous record of coincident failure) or beyond the scope of this paper, the basic physics of direct triggering must be mentioned as a reason for why these events, and indeed most earthquakes, are not expected to be time-predictable (e.g. Brodsky et al., 2020). Triggering is also clearly important in this specific case because the total moment is released by a pair of events, not a single one.

Otherwise, there are a few minor typos or errors that I expect will be caught in copyediting. The preferred spelling of the country hosting the doublet is now Türkiye, and that has been used widely in the research related to these earthquakes.

Reviewer #2 (Remarks to the Author):

The manuscript was substantially changed, eliminating the author's slip distribution and Coulomb stress changes analysis, in order to focus on the perhaps more relevant possibility of the exhibition of slip-predictable behavior of the fault segments that ruptured during the 2023 Turkey earthquakes.

I find that the reviewers' observations for the previous version of the manuscript were mainly addressed by the authors (with the exception of those related to the eliminated analyses, which have no effect anymore).

The main results regarding the slip-predictability as a lower bound for the 2023 earthquakes appear to be robust throughout the variations of the possible last ruptures for the different segments. The authors made many simplifications and assumptions during their analysis, but they clearly state them and explain their basis, also pointing out the limitations of the results.

Figure 1 has very bad resolution, particularly the small panels. I assume the blue and red circles on the large panel are the epicenters of the earthquakes, but it should be stated on the caption. Also, the Hellenic and Cyprian arcs should be located on the figure, since they are mentioned in the main text.

Figure S2 needs the references for the slip rates shown (or at list a reference to the table where those references are listed).

In line 110, I believe the authors meant to write: "equivalent to an 8-moment magnitude event".

I think the manuscript is ready for publication.

Reviewer #3 (Remarks to the Author):

I think the authors have done a thorough job of addressing my comments (as well as those of the other reviewers). The revised manuscript is more focused on addressing a clear research question, it has eliminated many of the problematic analytical issues in the original version, and the figures are improved. I like the direct approach of averaging different models for a unified coseismic slip value along each segment; the authors have effectively communicated this analysis. I have some comments and questions below that I think could help address some minor remaining issues with the manuscript.

Comments

ll. 38-68: I think these two paragraphs might need a little reorganization. Right now, they describe observations (ll. 38-46), then interpret the observations (ll. 46-57), then go back to similar observations (ll. 59-68). I would put the observations together, then have a separate paragraph explaining why certain events might be more or less in line with a particular

recurrence model.

I. 39: I would add “the time- or slip-predictable model” into the phrase “neither model is universally applicable”, as there are other, more sophisticated, recurrence models

I. 45: What exactly constitutes a “large seismic catalog”? Are these regional earthquakes? Fault-specific? I think this needs a couple more words of description.

I. 55: Probably the most salient point in this discussion is our ability to observe the physical state of a fault. I would emphasize this more. At a large scale, we can better observe bulk properties of a major fault system. Once the scale is smaller, we simply cannot resolve fault features anymore.

II. 81-83: The argument as stated here sounds a bit too much like circular logic: large earthquake size (referring specifically to magnitude? rupture length?) produces large earthquake slip, which of course produces large earthquakes. One (less circular) argument that could be made is that the extreme rupture length allowed the 2023 earthquakes to release more of the slip deficit accumulated not only since the previous large events, but also any slip deficit that those events did not release (a similar interpretation has been made for subduction earthquakes: Herman et al., 2018, Fig. 8).

II. 95-96: Specify westward motion of Anatolia with respect to Eurasia (or some other reference frame)

I. 124: Rather than “inherent uncertainties” I would call these differences in slip models “variations”

II. 145: Strictly speaking, the authors’ approach does not reduce uncertainties. I might remove the phrase “To further reduce uncertainties...interseismic deformation” and start the paragraph with “We calculate the average...”

II. 204f: It would be interesting and probably useful to compare fault slip rate estimates with the plate motion vector along the EAF system

I. 205: The authors repeatedly use “interseismic slip rates” throughout this section but I do not believe this is exactly what they mean. By definition, the faults are not slipping if they are locked and accumulating slip deficit. I believe the authors are referring to relative motion across the locked fault (not exactly plate motion for the GCFS, but akin to that), or perhaps the rate of motion across the deeper, creeping section of the fault.

II. 291-292: Please explicitly list out the fault segments corresponding to each penultimate event (as in the first half of this sentence)

II. 299-301: Unless the preceding event did not release all accumulated slip deficit, leaving some residually on the fault

II. 317f: It might be interesting to calculate how many years of slip deficit the 2023 event on the EAF released, and compare this to the earthquake history

I. 320: Where does the stress drop estimate come from? Reference?

II. 332-334: I think it is very important to distinguish between stress drop and moment release in this discussion. They are not the same thing (e.g., Almeida et al., 2018; Herman et al., 2018), and the difference may be relevant to the authors’ discussion.

II. 347-348: Is the difference with the alternative scenarios that the penultimate earthquakes happened more recently? This should be stated here, even if the details are in the Supplement.

II. 357-394: This analysis of time-varying slip rates seems like good supplementary material

I. 436: Should “ingestion” be “input”?

References

Almeida, R. et al. (2018). Can the updip limit of frictional locking on megathrusts be detected geodetically? Quantifying the effect of stress shadows on near-trench coupling. *Geophysical Research Letters*, 45(10), 4754–4763.

Herman, M.W. et al. (2018). The accumulation of slip deficit in subduction zones in the absence of mechanical coupling: Implications for the behavior of megathrust earthquakes. *Journal of Geophysical Research: Solid Earth*, 123(9), 8260–8278.

Response to Reviewers for "Were the 2023 M_w 7.8-7.7 Turkey Earthquakes Slip-Predictable?" (COMMSENV-23-0652A)

We thank the reviewers for their thoughtful comments. We incorporated them in the revised manuscript. Below we provide our detailed responses for all items in the respective reviews. Original comments are in **black** and our responses are in **blue**.

General comments:

Here we provide a summary of the general changes made to the manuscript in accordance with reviewer comments; many associated details are elaborated upon in the subsequent direct reviewer response sections.

Following suggestions of the reviewers, we have restructured the manuscript to focus on testing the slip-predictability of the 2023 Kahramanmaraş earthquake doublet. The main research question is formulated in the title of the manuscript. Note that we updated the moment magnitude of the 2nd event based on new estimates (e.g., Jia et al., 2023). We apologize for the long delay in resubmission due to the substantial nature of the revisions, but we feel our conclusions are now more robust and the manuscript overall has significantly strengthened.

As numerous studies on the 2023 Kahramanmaraş sequence have been published since the submission of our original manuscript, we have opted to remove the sections pertaining to coseismic slip modeling and Coulomb stress changes. We now include interseismic slip rates and coseismic slip estimates from multiple published studies. Thus, the presented results span a range of independent expert estimates, allowing us to better quantify uncertainties. We describe in detail the procedure by which we integrate the different slip models and consider the quantitative impacts of several key assumptions made in the newly extended discussion section.

We also have added a new section which provides a more complete overview of both the qualitative and quantitative information regarding historic ruptures along the East Anatolian Fault system, as well as their associated uncertainties. We consider multiple scenarios of penultimate events, as in some cases there is ambiguity regarding the size, extent, and location of historic earthquakes.

We confirm our original finding that the coseismic slip and interseismic slip deficit are strongly correlated, but we also find evidence that some of the segments exhibit slip overshoot; slip-predictability holds essentially as a lower-bound. This suggests that major earthquakes that rupture individual segments of the East Anatolian Fault produce incomplete stress drops, while the cascading multi-segment ruptures appear to relieve at least some fraction of the residual stress left behind by prior events. The new results presented in the revised manuscript highlight important lessons from the 2023 Kahramanmaraş earthquake sequence and provide new perspective on the recurrence behavior of large strike-slip earthquakes.

Reviewer #1 comments:

This manuscript uses regional GPS observations to invert for the slip model of the 2023 Türkiye earthquake sequence. With the slip models, it also presents Coulomb stress changes and compares the inverted amount of slip to the historical earthquake record, supporting a hypothesis of slip-predictable events in the region. Each of these results is derived from well-established methods and appropriate observational constraints. The first two, the details of event displacement and consequent regional static stress changes, are consistent with other studies of the earthquakes. The latter is a more surprising result since few, if any, major faults have been shown to be slip-predictable.

The first two major results, therefore, are consistent with and incremental to closely related studies. The approach has been used on hundreds of moderate to large earthquakes globally. As a result, a study like this would normally be published in topical or regional journals. The primary justification for publication in Nature Earth and Environment Communications is the rapid reduction of data combined with the societal impact of the earthquake sequence. Because these events had disproportionate impacts to human communities, they are of particular and timely scientific interest. The contributions of the slip model and stress change results to general earthquake mechanics, seismology, or tectonics are incremental.

The third major result, of slip-predictability, may have broader scientific impacts. Because fault systems are complex and highly nonlinear, slip-predictable seismicity over multiple earthquake cycles has seldom been observed in the seismic record. Instead, most theoretical models and observational constraints indicate that the amount of elastic slip potential stored on a fault system since the previous rupture should not be used to determine the likely magnitude or timing of the next event. However, the East (EAF) and North Anatolian Fault (NAF) may differ from “typical” fault systems by having simpler behavior. For example, the NAF ruptured in a nearly-linear temporal and spatial sequence in the 20th century, providing the most famous example of earthquake triggering in seismology. If this structured behavior is the consequence of structural simplicity or a more linearized frictional law for the system, one might expect rupture behavior closer to the slip-predictable end member. This possibility is more scientifically novel, and perhaps worthy of more development and emphasis in the manuscript.

I would therefore suggest two editorial options for the manuscript. If the editor would like to get a paper out about the Türkiye events quickly because of their non-scientific impacts, the paper could be published in its current form with minor grammatical and style revisions. **If, on the other hand, the editor would prefer to see a manuscript with more potential for scientific impacts, the authors should be encouraged to incorporate more numerical simulation (or citations from prior work on fault systems with slip predictability) to expand on the possibility that the EAF and the NAF could be natural examples of the most simple bounding case for slip-predictability.**

We thank the reviewer for their positive comments. Because many well-constrained slip models have been published since the manuscript submission, we now use these published results, and focus on the hypothesis of slip-predictability. In the revised introduction, we provide a broader overview of fundamental models and past studies of earthquake recurrence. Similarly, the revised discussion section contains more thorough evaluation of the certainty of the results, as well as inferences as to what aspects of fault mechanics and rupture propagation contributed to the observed strain release during the 2023 earthquake doublet.

Reviewer #2 comments:

We thank the reviewer for their detailed comments on our manuscript. We have addressed all items, as described below, although we note that several comments refer to figures and sections of the original manuscript that were removed from the revised manuscript.

1. Line 39: Consider changing the title “Main” for “Introduction”.
Done.
2. Figure 1: Please consider the following suggestions to improve the clarity of the figure:
 - a. Panels c), d), and e): In general, use darker colors for the lines showing features on the maps (faults, borders, coasts).
We have darkened the color of line features.
 - b. Panel c): Use a darker shade of gray for the text within the figure. Use different tones of green for the aftershocks and the fault traces. Use different tones of red (or blue) for the faults that ruptured for each earthquake and their corresponding GPS displacements.
We have darkened the text color. We have changed aftershocks to black and faults to gray. The fault segments are color-coded (red/blue) by rupture. Since we have removed the coseismic modeling component of this study, we have omitted the coseismic GPS displacements.
 - c. Panel d): Add some explanation in the caption for the arrow(s). It’s hard to tell at first glance that the blue arrow is the large displacement that doesn’t fit in panel c), and all the other displacements are still on that map but they are tiny dots due to the scale (they are almost invisible).
We removed the respective panel from the revised manuscript.
3. Panels a) and b) are a very interesting way of showing the time series. These were completely new to me. If possible, consider adding a figure in the Supplementary materials, with a couple of examples of separate time series, showing the daily positions (and the positions at the rate they were acquired between the earthquakes) to appreciate better how the offsets were calculated. What was the acquisition rate during the interval between earthquakes?
The revised manuscript no longer uses coseismic GNSS data to derive the coseismic slip model, so the respective panels have been removed.
4. Slip distribution model: How was the maximum slip depth constrained for this model (fault width)? Was it limited by the fault width suggested by the aftershocks, or by the locking depth from the interseismic modeling?
We now use the coseismic slip model of Jia et al. (2023) that is based on much more data, as the “preferred model”.
5. Figure 2 b) & c): The units on top of the figures should be mm/yr instead of mm.
We have corrected this.
6. Lines 112-113, 534: What is the meaning of “The preferred model fits 99% of the input data”? I think a different quantitative measure of the misfit (or of the goodness of fit) might be more illustrative, such as rms errors or reduced χ^2 .
We have modified this sentence to instead state, “The preferred model provides a 99% reduction in the data variance.”
7. Line 113: It is written “Supplementary Figs. 8 and 9”. Is it “Supplementary Figs. 2 and 3”?

This sentence has been removed as we have removed the original coseismic modeling section.

8. Models of interseismic motion: How do the authors reconcile the differences on the locking depths obtained from the models for adjacent faults? Are these locking depths consistent with the fault width suggested by the aftershock distribution? Are these locking depths consistent with the maximum depth of the rupture (see comment 4) as modeled from GPS data?

We have performed additional analysis of the fault locking depths as we now incorporate the locking depth into our estimates of slip-per-segment (see new sub-section, "Evaluating coseismic slip from finite fault models").

We consider the depth distribution of aftershocks and historic seismicity, as well as the maximum depth of slip obtained from finite slip models of the coseismic rupture. We find that the 95% cutoff depth of seismicity tends to agree with the depth extent of coseismic slip.

9. Coulomb stress changes: The authors divide their predicted Coulomb stress changes in intervals starting on < 2 kPa. Are all the values shown on the plots positive, so that this first interval corresponds to 0 -2 kPa? Were there any faults for which a shadow zone was predicted (negative stress changes)?

We have removed the Coulomb stress analysis from the revised manuscript.

10. In the Discussion section, for the Slip model part, the authors make a comparison of surface fault slip extracted from orthophotos and predicted by their preferred model. There is no reference to data property regarding the photos (which agency provided them, acquisition credit, etc.). Are the offsets from orthophotos from a published source or were they measured as part of the present study? If it is the former, the citation is missing. If it is the latter, this was work done by the authors to validate their model, so I think it deserves at least a figure in the main manuscript (unless there are restrictions for the allowed number of figures). A couple of examples of these orthophotos in the Supplementary Material showing the displacements could be illustrative. Also, I would like to see a histogram of the residuals from this comparison in the Supplementary Material, it would give me a better sense of how closely the predictions match the observations. Is there a way to assess the uncertainty of these observations (and the model)?

Given that we are no longer including the original coseismic slip model, we have removed this portion of the discussion section.

11. Slip rate, Coulomb stress changes effects on future seismicity: It is not clear to me how the authors estimated the equivalent Coulomb stress increase corresponding to a given slip rate.

We have removed the Coulomb stress analysis.

12. Consider adding a Conclusion section to wrap up your results around the central idea of the manuscript as stated in the title. It is my impression that it is not emphasized enough.

We have added a conclusion section to summarize the main findings of this manuscript.

13. References: consider adding the doi to the references that have one.

Done.

14. Reference 8: The title of the article is missing.

We have corrected this.

15. Consider adding a second panel to Supplementary Figs. 2 and 3, showing the residuals from the model (observed minus calculated displacements), to show more clearly the performance of the model and also to see if there are any patterns in the residuals worth discussing.

These figures have been removed as we no longer include a new coseismic slip model.

16. Supplementary Figure 9: The rectangles with values that go with each vector should be placed in a way that they don't cover other arrows or important parts of the image.

We have removed this figure.

References

Jia, Z., Z. Y. Jin, M. Marchandon, T. Ulrich, A. Gabriel, W. Y. Fan, P. Shearer, X. Y. Zou, J. Rekoske, F. Bulut, et al. (2023). The complex dynamics of the 2023 Kahramanmaraş, Turkey, Mw 7.8-7.7 earthquake doublet, *Science* eadi0685, doi: 10.1126/science.adi0685.

Reviewer #3 comments:

We appreciate the reviewer's thoughtful and thorough comments. To address the main concerns brought up, we have performed major revisions and rewriting of the manuscript.

Major Comments

In my opinion, the authors are missing a main scientific question in the Introduction. They include a description of the tectonics, the recent earthquake sequence, and their analytical approaches/results. But I think their paper would be strengthened by setting up a research question that their work answers. The manuscript is also missing some references. The authors are not the first to look at the coseismic displacements for these events, generate slip models for these earthquakes, interpret the main event as starting on a smaller, non-EAF fault, or calculate Coulomb stress changes. I highly recommend that the authors look at some of the other studies that have been done already and examine their results in light of those other analyses.

Following the reviewer's remarks, we have substantially reworked the manuscript and now focus on evaluating the hypothesis of slip-predictability and insights to earthquake recurrence. We have extended the introduction, reframing the study in the context of earthquake recurrence, and discussion, focusing more on this evaluation of slip-predictability and elaborating on implications from our findings for earthquake ruptures and recurrence along the EAF.

With regards the latter half of this comment, we ultimately decided to remove the coseismic modeling and Coulomb stress change analysis from the study due to the number of studies which have since performed similar analyses. While we still incorporate finite slip models in order to address the question whether the 2023 doublet earthquakes were slip-predictable, we now instead use a collection published slip models from other studies. In addition to these changes, the manuscript contains now greater number of references to related studies of the 2023 Kahramanmaraş doublet.

The authors make several assertions about their slip inversion results that may be true but appear to be unsupported by evidence. Specifically, they call their model "relatively low resolution" (l. 121), but do not define what this means or how they determine its resolution. They say the "slip distribution is necessarily smoothed" (l. 121) but provide no reasoning for applying smoothing constraints or how much or by what means the model is smoothed. Finally, they claim that "the spatially averaged slip...is well constrained" (ll. 122-123), but provide no uncertainty estimates or evidence for robustness. Even if these show up later in the Methods section, it is important for the authors to provide some quantitative support to these statements here.

As we have removed the coseismic modeling section, these comments address text which is no longer included in the revised manuscript. However, we do note that some of these concerns are addressed via our more generalized approach to estimating the average slip along each segment of the 2023 ruptures. Rather than computing values from the original finite slip model included in the original manuscript, we now utilize a suite of models obtained from the literature. We estimate the average slip from each model in a consistent manner (see sub-section, "Evaluating coseismic slip from finite fault models")

I have a couple of concerns about the authors' interseismic and coseismic inversion approaches. I am not sure the authors provide enough information in all cases that I could reproduce their results, and I am very familiar with these types of inversions. For example, the assumptions underlying the "buried dislocation model" (l. 490) should be described, perhaps with a formula.

We have added the statement, "...wherein we assume a homogeneous elastic medium and that fault deformation is purely fault parallel and due to strike-slip motion" and added a citation to Savage & Buford (1973) and Segall (2010).

How long is a fault before it is “adequately approximated by a 2-D antiplane strain model” (ll. 491- 492)? We clarify that the fault segments should be at least several times longer than their inferred locking depths.

The profile explanation (ll. 492-493) does not explain the selected width of velocities around the profile or whether profiles intersect multiple faults.

We have re-written this sentence to state, “For each segment, we defined a fault-perpendicular profile with width of at least 100 km to ensure that the asymptotic nature of the fault-parallel velocities is adequately captured, and extracted the GPS velocity data along the profile.”

Finally, the description of uncertainty analysis is too full of jargon and does not explain the uncertainty estimation approach well (ll. 494-497). Similar gaps are in the coseismic modeling description.

We now more clearly define and describe the bootstrap procedure used, please see lines 759-766 of the revised manuscript.

Another concern I have about the interseismic modeling is whether it is done correctly. In Supp. Fig. 4, the profile crosses the EAF and the Çardak Fault to the north, yet I see no jump in velocities across the Çardak Fault as shown in Supp. Fig. 7. I am not sure this is completely explained by the projection of velocities onto the fault. Then, in Supp. Fig. 7, I do not understand why the authors cut the profile off at the Pazarcık segment of the EAF. In addition, they show a 9 mm/yr difference across the Pazarcık segment, half of which is completely absent here. I think the authors need to make sure this analysis is done correctly.

The apparent discrepancy between Supp. Figs. 4 & 7 is due to a combination of fault orientation (and thus projection), swath size, and difference in signal amplitude. The swath used for the Çardak fault is intentionally cut off at ~60 km to the south before the to avoid fitting the signal due to the main strand of the EAF (Pazarcık), which exists residually when projected into the fault-parallel azimuth associated with the Çardak fault.

Conversely, the velocity signal of the Çardak fault is slightly visible in the Supp. Figs. 4 around the 60 km mark. However, due to the lower slip rate and difference in fault strike (i.e., stations on opposite sides of the Çardak fault are re-ordered during the projection onto the Pazarcık strike, resulting in non-monotonic changes in the deformation signal away from the swath origin), it appears as a relatively 1-2 mm deviation from the primary “arctan” signal due to the Pazarcık. We acknowledge that the choice to have an asymmetric profile for the Çardak fault and not formally accounting for the Pazarcık fault’s signal (or vice-versa) may bias the estimated slip rates from this analysis. This is the reasoning for comparing the 2D profile inversion results with the 3D model. In the former, we assume that all deformation within a profile is due to a single fault in order to leverage the fault-parallel projection of the GPS velocities. In the latter we simultaneously account for the deformation of all modeled faults. However, there are additional uncertainties associated with more complex fault-perpendicular motions that are unmodeled in the full velocity field. In the end, we find that slip rate estimates along the Çardak fault from both models are in good agreement, suggesting that the slip rate is robust to the modeling assumptions made.

The slip-predictable earthquake model states that each earthquake releases all of the accumulated slip deficit. But that also assumes that we know what the accumulated slip deficit is (not just the slip deficit rate). I suspect that M 7.5+ earthquakes in strike-slip systems may locally release most of the accumulated slip deficit and essentially reset the clock. However, I am less confident that M 7.0 (e.g., l. 252) or smaller events do the same. They will leave slip deficit on the fault. Related to this, the authors do not provide magnitudes for all of the historical events they discuss, which to me is a major omission.

A significant update to the manuscript is the addition of the “Historic ruptures along the East Anatolian fault system” section. In this section, we discuss the likelihood of each most recent large (M7+)

earthquake on each fault segment and provide basic quantitative information when available, (i.e. magnitude estimates). We also discuss additional possible rupture scenarios in the supporting information.

We have also added additional discussion and clarification with regards to the former point. First, our results now reflect slip-predictability along the Çardak and Göksun segments whereas the main strand of the EAF appears to have relieved slip deficit in excess of that accumulated since the preceding earthquakes. This implies, as referenced above, that some existing slip deficit was “leftover” from past earthquakes, and thus, unsurprisingly, that there is a degree of complexity in large earthquake recurrence on the EAF. While this result is less peculiar than then wholesale slip-predictability of both events, there is still valuable insight from this observation. This includes the fact that we still observe a strong correspondence between the slip deficits and moment release (Fig. 5), so the ultimate size scaled with the observed strain accumulation. While from a seismic cycle perspective, the reciprocity between events which relieve more or less strain than had accumulated since the previous one is not surprising, in a practical hazard assessment sense the documentation of strain overshoot is important. In short, our results suggest that a larger slip deficit is likely to imply a large earthquake; however, the upper limit to the strain release is not necessarily limited to the accumulated slip deficit, but rather the nature of the previous (or several previous, perhaps) earthquake(s) along the fault. We elaborate on these points in the revised discussion section.

Several of the figures probably need substantial improvement before publication. Fig 1a and 1b appear to be time series, but I find these almost unreadable. Arrows for displacements should not go off the map, and faults should be more distinguishable from displacement observations in Fig 1c. Why does Fig. 1d need to show the displacements at all? Lastly for Fig. 1, the authors should specifically refer to these subpanels in the manuscript.

For Fig. 2, velocities derived from GPS are not “slip rate,” so these labels should be changed. We have changed the labels to “velocity (mm/yr)”.

“Latitude” is spelled incorrectly. We have corrected this.

This figure is not referred to in the manuscript until the Methods, so it should come later. We have added appropriate references (first instance 104).

The basemap for Fig. 3 is too low resolution. We have removed this figure.

The authors should include negative values for Coulomb stress changes. The Coulomb stress analysis has been removed.

Minor Comments

The names of the major faults (EAF, NAF) are well established and known by the geological community. The segments and other faults are not as well known and therefore are not particularly meaningful to a general audience. If the authors choose to use these segment names, I think they should define the segment locations in the Introduction with a companion map. I also think it would be helpful to indicate whether these are broadly accepted segment labels or if the authors are naming the segments here for the first time.

The segments described in the new section, “The 2023 M_w 7.8-7.7 Kahramanmaraş earthquake doublet,” where we specify that we are using the segment extents and names given by Duman & Emre (2013). Figure 1C is a map illustrating the spatial extent of the segments.

l. 26, l. 44, l. 56: The term “doublet” carries a lot of implicit meaning, and different people use the term differently. I suggest defining the specific usage here or using another term. It becomes particularly confusing when talking about the doublet being “the largest seismic event in the region” (l. 57). Are the authors referring to combined seismic moment release? The initial, larger earthquake?

We now state, “The remarkable 2023 MW 7.8-7.7 Kahramanmaraş earthquakes in Turkey, which are considered to be doublet due to their similar large magnitudes and occurrence in rapid succession (~9 hrs apart) in Line 70. We have also rephrased the comparison to the 1668 earthquake, now stating, “The latter event alone would have been the largest earthquake in the region since the devastating August 17, 1668 M 7.9 earthquake on the NAF42; the total moment release of the doublet, equivalent to moment a magnitude 8 event¹⁷, may have been the largest ever documented in Turkey” in Line 108.

ll. 73: Aftershocks are not mentioned at all in the Introduction, nor are they introduced at the beginning of this section. I think the authors should motivate why aftershocks are relevant to discuss here instead of leaving that unsaid.

We now use the aftershock dataset to evaluate locking depths along the segments of the EAF that ruptured in 2023, as mentioned in the section “Coseismic rupture along the East Anatolian fault.” We have removed the old “Aftershock distribution” sub-section, but retained the additional comments in the supporting information.

I am not familiar with what information should be in the Methods section at the end of the manuscript and what material should be in the Supplementary files. My instinct is that if a method is critical for an argument in the paper, it should be in the main body, not a supplement. Then, the authors should refer to the Methods section explicitly in the main body of their paper (e.g., when talking about the GPS displacements in ll. 96-97 or the slip model inversions in ll. 110-111). Otherwise, I feel like the manuscript is missing a lot of information that is probably included elsewhere.

We have added additional references to the Methods section for the slip rate inversions and locking depth estimates.

I had difficulty following the figures because the numbering is out of order in the manuscript. I also found no reference to Figure 2. Please correct this.

The manuscript now has in-text references to Figure 2 (first occurrence on Line 104).

It is unclear to me why the authors determine the background stress field (ll. 165-168). I do not see how they use these results in their analysis. Calculating stress changes generated by the earthquakes on faults with known geometry usually precludes the need to calculate background stresses (which is generally done to determine the optimal fault orientation when fault geometries are unknown). Also, I think it might be worthwhile for the authors to show faults with negative Coulomb stress changes, because earthquakes on these structures might be delayed.

We have removed the stress change analysis from the manuscript.

The “Discussion” section in ll. 201-226 is not much of a discussion at all. I think if the authors expanded this section into a fuller comparison between their analyses and other analyses, this could be a valuable element of the discussion. But as it stands, this section reads more like the end of a Results section than a useful discussion of the implications of the analysis.

As part of the major revisions to the manuscript, we have significantly expanded the discussion section (now titled, “Results and Discussion”). There is now a more thorough discussion of the uncertainties associated with the primary findings related to recurrence behavior, as well as implications for fault and rupture mechanics based on our findings.

Detailed Comments

l. 27: I do not feel it is appropriate to combine the length of the two fault ruptures, since they are clearly different geological structures that had distinct earthquakes

We have removed this statement

l. 41: The “entire southern section” is too ambiguous a description. Please define the extent of the fault that ruptured and/or refer to the region on a figure.

We have changed this to specifically state the Amanos, Pazarcık, and Erkenek segments, as defined in Duman & Emre (2013).

ll. 51-55: I think this sentence is too long and should be broken into 2 for clarity. I also think that the Hellenic subduction zone, while relevant to regional tectonics, might not be particularly relevant to these earthquakes near the EAF.

We have separated this sentence into two and slightly reworded for clarity. We retained reference to the Hellenic arc (and Cyprian arc) for completeness regarding the description of the regional tectonics, as well as the fact that the Cyprian arc is now included in the 3D interseismic deformation model.

l. 56: What region are the authors referring to specifically for the frequency of M7+ earthquakes? Turkey as a country? The Anatolian Plate boundaries?

This has been changed to refer specifically to the country of Turkey.

l. 62: I assume the continuous GPS data are daily positions, but this should be stated explicitly

We no longer include the coseismic GPS measurements in the revised manuscript.

l. 66: I would disagree with the characterization of interseismic deformation measured with GPS as “long term”

The paragraph this comment refers to has been re-written; however, we include more detailed commentary on the degree to which modern GPS measurements may reflect long-term strain accumulation in the discussion section. We also compare geodetic and geologic slip rates (Fig. S2), which shows that there is a noticeable 2-3 mm/yr discrepancy. We additionally discuss the possible implications of this in the discussion.

l. 84: Start a new paragraph with “The aftershock distribution...”

The aftershock section has been removed from the main text.

l. 96: The term “processed” does not capture the information I am most interested in from the GPS. I recommend the authors explicitly indicate that they determined coseismic displacements.

We have removed the coseismic slip modeling component of the study from the manuscript

l. 108: Please define “most reliable.” Are the authors referring to the sampling rate? The data distribution? The ability to model coseismic deformation from the dataset?

We have removed the coseismic slip modeling component of the study from the manuscript

ll. 189: I found it a little unusual to jump between fractions of MPa and multiple KPa for the Coulomb stress changes. I would prefer to stick with one unit and describe all of the results with that unit.

We have removed the Coulomb stress change analysis from the manuscript.

l. 221: Reference for dynamic stress changes being larger than static?

We have removed the Coulomb stress change analysis from the manuscript.

ll. 222-223: What is the relevance of a lack of foreshock activity to the Coulomb stress change results?

We have removed the Coulomb stress change analysis from the manuscript.

l. 224: The word “governed” is too non-specific

We have removed the Coulomb stress change analysis from the manuscript.

References

Duman, T.Y. & Emre, Ö. The East Anatolian Fault: geometry, segmentation and jog characteristics. *Geol. Soc., Lond., Spec. Publ.* 372,495–529 (2013).

Savage, J. C. & Burford, R. O. Geodetic determination of relative plate motion in central California. *J. Geophys. Res.* 78, 832–845 (1973).

Segall, P. *Earthquake and Volcano Deformation*. (Princeton University Press, Princeton, NJ, 2010).

Response to Reviewers for "The 2023 M_w 7.8-7.6 Turkey Earthquakes follow the slip-predictable model" (COMMSENV-23-0652A)

We thank the reviewers for thoughtful comments and suggestions. We incorporated them in the revised manuscript. Please see below our detailed responses. Reviewers' comments are in **black** and our responses are in **blue**.

Reviewer #1

It appears that this manuscript has already gone through at least one round of reviews and revisions, so overall it is nearly ready for publication. I can mostly contribute by drawing the authors' attention to a few papers that offer an explanation for the variability of moment magnitudes and their relationship to stored elastic energy or apparent slip overshoot. Specifically, Feldl and Bilham (2006) and a few others have pointed out that larger magnitude events may actually release mechanical potential energy (elastic or elastio-plastic) from larger volumes than smaller events, where the fault length scaling varies with L , the moment with L^2 , but the "reservoir" of stored energy with L^3 . It is therefore not entirely straightforward to convert the accumulated tectonic strain into slip potential because the former value is sensitive to the length of the baseline over which that strain is measured but larger events might actually discharge potential from a longer baseline or, actually, a larger volume. This effect is expected to produce the observations that the authors note: that larger events might look like they are overshooting the stored strain because that storage is not measured over a long enough baseline. This effect appears to be more important in continental settings, where strain is stored on faults and in volumes, where faults are less mature, and where there is a lot of kinematic complexity. All of these conditions apply to the East Anatolian Fault, so this study is a nice example of both the relevance of slip-predictability and the potential for nonlinear modulation of that slip-predictability. In my opinion, this added nuance could be enhanced with just a sentence or two and a citation or two, but it would increase the impact of the work.

We appreciate the Reviewer's suggestions. The model presented by Feldl and Bilham (2006) is based on the observation of interseismic strain accumulation to the north of the Main Thrust, where no large active faults are known to accommodate the accumulated strain over time. The argument of Feldl and Bilham is that occasional large earthquakes may produce slip on the decollement that extends much further to the north than observed so far, in order to relieve the accumulated strain. This model is not readily applicable to the Kahramanmaraş earthquakes (and strike-slip faulting in general). In the case of the East Anatolian fault system and other strike-slip faults further to the west, there is no "excess strain" that is unaccounted for given the mapped fault distribution and measured interseismic velocity field.

One other effect that is the target of a lot of recent attention is that of synchronization or entrainment of triggered cascades, leading to effects like supercycles in synthetic catalogs (Field et al., 2017; Field et al., 2022; Milner et al., 2022; Brodsky and van der Elst, 2014; Van der Elst and Brodsky, 2010). This probably also warrants a sentence or two, since the event is a doublet consisting of events expected to have very different characteristic timescales. Even if multicycle temporal alignment is considered unlikely (i.e. this set of faults does not have a previous record of coincident failure) or beyond the scope of this paper, the basic physics of direct triggering must be mentioned as a reason for why these events, and indeed most earthquakes, are not expected to be time-predictable (e.g. Brodsky et al., 2020). Triggering is also clearly important in this specific case because the total moment is released by a pair of events, not a single one.

We have added several sentences regarding the triggering of the second event in the doublet (see paragraph beginning at line 423 in the finalized manuscript). In general, we feel that the updated discussion section now better relates the complex dynamic effects which have been documented in

modeling studies to higher-level concepts which may ultimately govern the magnitude of large earthquakes.

Otherwise, there are a few minor typos or errors that I expect will be caught in copyediting. The preferred spelling of the country hosting the doublet is now Türkiye, and that has been used widely in the research related to these earthquakes.

We have now use Türkiye instead of Turkey throughout the manuscript, as suggested by the Reviewer.

Reviewer #2

Figure 1 has very bad resolution, particularly the small panels. I assume the blue and red circles on the large panel are the epicenters of the earthquakes, but it should be stated on the caption. Also, the Hellenic and Cyprian arcs should be located on the figure, since they are mentioned in the main text.

We have specified that the circles denote the earthquake epicenters and added the traces for the Hellenic and Cyprian arcs. We also make several minor graphical edits to improve the visual clarity of the figure. The resolution of the figure in the manuscript file now has improved resolution.

Figure S2 needs the references for the slip rates shown (or at list a reference to the table where those references are listed).

We have added a reference call to Table S2, which lists all the slip rates used and their associated publications.

In line 110, I believe the authors meant to write: “equivalent to an 8-moment magnitude event”.

We have corrected this sentence to, “the total moment release of the doublet, equivalent to a M_w 7.95 event¹⁷...”

Reviewer #3 (Remarks to the Author):

ll. 38-68: I think these two paragraphs might need a little reorganization. Right now, they describe observations (ll. 38-46), then interpret the observations (ll. 46-57), then go back to similar observations (ll. 59-68). I would put the observations together, then have a separate paragraph explaining why certain events might be more or less in line with a particular recurrence model.

We have reworded this section. We have grouped discussion of different observations and their compatibility (or lack thereof) with recurrence models in the first paragraph, with the second paragraph considering broader interpretations in the second paragraph.

l. 39: I would add “the time- or slip-predictable model” into the phrase “neither model is universally applicable”, as there are other, more sophisticated, recurrence models

We have rephrased this to say, “...suggesting that neither the time- nor the slip-predictable model is universally applicable...”

l. 45: What exactly constitutes a “large seismic catalog”? Are these regional earthquakes? Fault-specific? I think this needs a couple more words of description.

We have clarified we are referring specifically to modern (i.e. instrumental) regional catalogs, and cite a study that considered seismicity in Italy as an example.

l. 55: Probably the most salient point in this discussion is our ability to observe the physical state of a fault. I would emphasize this more. At a large scale, we can better observe bulk properties of a major fault system. Once the scale is smaller, we simply cannot resolve fault features anymore.

We have added several additional phrases to the revised version of this paragraph to emphasize this point.

ll. 81-83: The argument as stated here sounds a bit too much like circular logic: large earthquake size (referring specifically to magnitude? rupture length?) produces large earthquake slip, which of course produces large earthquakes. One (less circular) argument that could be made is that the extreme rupture length allowed the 2023 earthquakes to release more of the slip deficit accumulated not only since the previous large events, but also any slip deficit that those events did not release (a similar interpretation has been made for subduction earthquakes: Herman et al., 2018, Fig. 8).

We have reworked this sentence and mention the rupture length as a major factor in facilitating the exceptionally high slip. This is elaborated on in the updated discussion section.

ll. 95-96: Specify westward motion of Anatolia with respect to Eurasia (or some other reference frame)

We have added, “with respect to Eurasia” to this sentence.

l. 124: Rather than “inherent uncertainties” I would call these differences in slip models “variations”

We have changed this sentence to use “variations.”

ll. 145: Strictly speaking, the authors' approach does not reduce uncertainties. I might remove the phrase "To further reduce uncertainties...interseismic deformation" and start the paragraph with "We calculate the average..."

We have removed the first part of this sentence as described.

ll. 204ff: It would be interesting and probably useful to compare fault slip rate estimates with the plate motion vector along the EAF system

It has been noted in the literature that the slip rate along the EAF decreases toward the southwest, as the fault becomes more mis-aligned with rigid plate motion. We have added a sentence stating this and added a relevant citation.

l. 205: The authors repeatedly use "interseismic slip rates" throughout this section but I do not believe this is exactly what they mean. By definition, the faults are not slipping if they are locked and accumulating slip deficit. I believe the authors are referring to relative motion across the locked fault (not exactly plate motion for the GCFS, but akin to that), or perhaps the rate of motion across the deeper, creeping section of the fault.

The Reviewer is correct in that an "interseismic slip rate" is a modeled parameter obtained by fitting the observed interseismic velocity field. We have replaced "interseismic slip rates" with "geodetic slip rates", a term that is commonly used in the literature, to avoid confusion.

ll. 291-292: Please explicitly list out the fault segments corresponding to each penultimate event (as in the first half of this sentence)

We have done this.

ll. 299-301: Unless the preceding event did not release all accumulated slip deficit, leaving some residually on the fault.

The referenced sentence states the strict definition of slip-predictability, by which the aforementioned scenario would not be considered to be slip-predictable. We have appended the "loose" definition of slip predictability, in where there is scaling between interseismic interval duration and earthquake size. This more generalized behavior is what we observe in the case of the of the M_w 7.8 event and is elaborated on in the following discussion.

ll. 317ff: It might be interesting to calculate how many years of slip deficit the 2023 event on the EAF released, and compare this to the earthquake history

We have added a table listing the slip rates, estimated slip deficits, predicted penultimate earthquake date ranges, and actual penultimate earthquake date ranges, as well as several sentences describing the results in the Discussion section (line 339 in the finalized manuscript).

I. 320: Where does the stress drop estimate come from? Reference?

We are not referencing particular stress drop values, but rather inferring that in order to leave stress on the fault to be released in slip overshoot during the 2023 doublet, the stress drop from the penultimate events was incomplete with respect to the fault strength.

II. 332-334: I think it is very important to distinguish between stress drop and moment release in this discussion. They are not the same thing (e.g., Almeida et al., 2018; Herman et al., 2018), and the difference may be relevant to the authors' discussion.

We agree that more nuanced treatment of these quantities is warranted. We have rephrased this particular sentence to focus on moment release but have also added significant additional discussion regarding moment and strain release per the suggestions of Reviewer #1. In general, we have moved away from discussion of stress drop in the updated manuscript, instead focusing on slip-deficit and moment release.

II. 347-348: Is the difference with the alternative scenarios that the penultimate earthquakes happened more recently? This should be stated here, even if the details are in the Supplement.

We have added brief summaries for each alternative scenario considered.

II. 357-394: This analysis of time-varying slip rates seems like good supplementary material

We have moved the original paragraphs to the supplement and inserted an abbreviated summary.

I. 436: Should "ingestion" be "input"?

We have changed "ingestion" to "analysis".

Review of: The 2023 M_w 7.8-7.6 Turkey Earthquakes follow the slip-predictable model

This manuscript presents the integration of various analysis to study the February 2023 earthquake doublet. Time series analysis of data from a network of continuous GPS stations that registered the earthquake allowed the estimation of coseismic displacements at the GPS sites for each of the earthquakes. These displacements were then inverted to obtain a fault slip model for which the various ruptured segments were defined by the aftershock distribution. The preferred model was validated by comparing surface displacements along the rupture measured from orthophotos with model predictions at the same locations. For each one of the segments that intervened in the earthquakes, the authors modeled the interseismic slip rate and locking depths using a published large scale velocity field. The slip rate estimates and the average coseismic slip allowed to bracket the time required to accumulate the appropriate slip deficit, which was in good agreement with the dates of the last historical earthquakes known for each segment. This provided a second validation of their slip distribution model. This result suggests that the doublet was consistent with a slip-predictable model for this mature fault system, rather than a characteristic earthquake or time predictable model. The coseismic slip model was also used to estimate the Coulomb stress changes imposed by each of the earthquakes on nearby faults. The second main shock and the aftershocks of both earthquakes were consistent with faults with increased stress. Moreover, the combination of the two earthquakes produced large Coulomb stress changes on several faults, which likely are now closer to failure.

The manuscript is very well written and is a pleasure to read through. I found the results very compelling. If not for a few details, the paper could be ready for publication as is. However, I noted a few places where the discussion could be refined and a few figures could be improved slightly.

Find my comments below.

Apart from these minor comments, please accept my congratulations on a very interesting paper.

Comments:

1. Line 39: Consider changing the title “Main” for “Introduction”.
2. Figure 1: Please consider the following suggestions to improve the clarity of the figure:
 - a. Panels c), d), and e): In general, use darker colors for the lines showing features on the maps (faults, borders, coasts).
 - b. Panel c): Use a darker shade of gray for the text within the figure. Use different tones of green for the aftershocks and the fault traces. Use different tones of red (or blue) for the faults that ruptured for each earthquake and their corresponding GPS displacements.
 - c. Panel d): Add some explanation in the caption for the arrow(s). It’s hard to tell at first glance that the blue arrow is the large displacement that doesn’t fit in panel c), and all the other displacements are still on that map but they are tiny dots due to the scale (they are almost invisible).
3. Panels a) and b) are a very interesting way of showing the time series. These were completely new to me. If possible, consider adding a figure in the Supplementary materials, with a couple of examples

of separate time series, showing the daily positions (and the positions at the rate they were acquired between the earthquakes) to appreciate better how the offsets were calculated. What was the acquisition rate during the interval between earthquakes?

4. Slip distribution model: How was the maximum slip depth constrained for this model (fault width)? Was it limited by the fault width suggested by the aftershocks, or by the locking depth from the interseismic modeling?
5. Figure 2 b) & c): The units on top of the figures should be mm/yr instead of mm.
6. Lines 112-113, 534: What is the meaning of “The preferred model fits 99% of the input data”? I think a different quantitative measure of the misfit (or of the goodness of fit) might be more illustrative, such as rms errors or reduced χ^2 .
7. Line 113: It is written “Supplementary Figs. 8 and 9”. Is it “Supplementary Figs. 2 and 3”?
8. Models of interseismic motion: How do the authors reconcile the differences on the locking depths obtained from the models for adjacent faults? Are these locking depths consistent with the fault width suggested by the aftershock distribution? Are these locking depths consistent with the maximum depth of the rupture (see comment 4) as modeled from GPS data?
9. Coulomb stress changes: The authors divide their predicted Coulomb stress changes in intervals starting on < 2 kPa. Are all the values shown on the plots positive, so that this first interval corresponds to 0 -2 kPa? Were there any faults for which a shadow zone was predicted (negative stress changes)?
10. In the Discussion section, for the Slip model part, the authors make a comparison of surface fault slip extracted from orthophotos and predicted by their preferred model. There is no reference to data property regarding the photos (which agency provided them, acquisition credit, etc.). Are the offsets from orthophotos from a published source or were they measured as part of the present study? If it is the former, the citation is missing. If it is the latter, this was work done by the authors to validate their model, so I think it deserves at least a figure in the main manuscript (unless there are restrictions for the allowed number of figures). A couple of examples of these orthophotos in the Supplementary Material showing the displacements could be illustrative. Also, I would like to see a histogram of the residuals from this comparison in the Supplementary Material, it would give me a better sense of how closely the predictions match the observations. Is there a way to assess the uncertainty of these observations (and the model)?
11. Slip rate, Coulomb stress changes effects on future seismicity: It is not clear to me how the authors estimated the equivalent Coulomb stress increase corresponding to a given slip rate.
12. Consider adding a Conclusion section to wrap up your results around the central idea of the manuscript as stated in the title. It is my impression that it is not emphasized enough.
13. References: consider adding the doi to the references that have one.

14. Reference 8: The title of the article is missing.
15. Consider adding a second panel to Supplementary Figs. 2 and 3, showing the residuals from the model (observed minus calculated displacements), to show more clearly the performance of the model and also to see if there are any patterns in the residuals worth discussing.
16. Supplementary Figure 9: The rectangles with values that go with each vector should be placed in a way that they don't cover other arrows or important parts of the image.